# Recent Advances in Inverted Perovskite Solar Cells: Designing and Fabrication

**DOI:** 10.3390/ijms231911792

**Published:** 2022-10-04

**Authors:** Jiayan Yang, Xingrui Luo, Yankai Zhou, Yingying Li, Qingqing Qiu, Tengfeng Xie

**Affiliations:** 1Engineering Research Center for Hydrogen Energy Materials and Devices, College of Rare Earths, Jiangxi University of Science and Technology, Ganzhou 341000, China; 2Faculty of Materials Metallurgy and Chemistry, Jiangxi University of Science and Technology, Ganzhou 341000, China; 3College of Chemistry, Jilin University, Changchun 130012, China

**Keywords:** inverted PSCs, charge transport materials, all−inorganic inverted device

## Abstract

Inverted perovskite solar cells (PSCs) have been extensively studied by reason of their negligible hysteresis effect, easy fabrication, flexible PSCs and good stability. The certified photoelectric conversion efficiency (PCE) achieved 23.5% owing to the formed lead−sulfur (Pb−S) bonds through the surface sulfidation process of perovskite film, which gradually approaches the performance of traditional upright structure PSCs and indicates their industrial application potential. However, the fabricated devices are severely affected by moisture, high temperature and ultraviolet light due to the application of organic materials. Depending on nitrogen, cost of protection may increase, especially for the industrial production in the future. In addition, the inverted PSCs are found with a series of issues compared with the traditional upright PSCs, such as nonradiative recombination of carriers, inferior stability and costly charge transport materials. Thus, the development of inverted PSCs is systematically reviewed in this paper. The design and fabrication of charge transport materials and perovskite materials, enhancement strategies (e.g., interface modification and doping) and the development of all−inorganic inverted devices are discussed to present the indicator for development of efficient and stable inverted PSCs.

## 1. Introduction

At present, humanity is faced with serious energy and environmental issues [1,2]. Even if timely energy conservation measures are taken, the world energy demand is expected to increase triple by 2050 [3]. In addition, the consumption of fossil fuels produces some nasty side effects, including air pollution and greenhouse effects [4,5]. Solar energy is a renewable clean energy [6,7]. Thus, solar cells, as the devices that convert sunlight into electricity directly, are considered as one of the efficient approaches to satisfying the appetite for clean energy in the future [8,9]. Until now, silicon (Si) solar cells possess the best photovoltaic efficiency and dominate the solar panel market [10,11]. Nevertheless, the popularity of Si solar cells is restricted by reason of their huge expense and serious pollution.

Perovskite solar cells (PSCs), as a novel type of solar cells, have received a lot of attention and research because of their wide light absorption range, high optical absorption coefficient and high carrier mobility of metal halide−based perovskite materials. With a few short years, PSCs have made a significant breakthrough, and the certified PCE has already achieved 25.5% [12]. To date, most PSCs have been constructed in the traditional upright structure. However, the traditional upright structure devices suffer from serious hysteresis effects and poor stability. Moreover, most of the traditional upright devices consist of a metal oxides electron−transport layer (ETL) with high sintering temperature, which leads to difficulty in achieving the development of flexible PSCs. Although numerous strategies, including surface passivation [13,14,15], interface engineering [16,17,18,19,20], doping [21,22,23,24,25], cross−linking of the perovskite film [26] and so on, have been developed to mitigate these issues, the PCE of PSCs is still well below the Shockley–Queisser limit efficiency based on theory (30.5%).

The typical inverted PSCs consist of hole−transport layer (HTL)/perovskite absorber layers (ABX_3_ (e.g., X = I^−^, Br^−^, Cl^−^))/ETL. The hysteresis effect of inverted PSCs can be significant reduced compared to that of traditional upright PSCs. Moreover, the inverted structure possesses good interface stability, fewer defect states between interfaces, and it is beneficial to prepare the flexible devices with low temperature process [27,28,29]. Although development of inverted PSCs is delayed, there is a lot of room for progress. Starting from 2013, Jeng et al. employed the PEDOT:PSS for the hole transport material (HTM), CH_3_NH_3_PbI_3_ (MAPbI_3_) for the perovskite absorber layer, C_60_ or C_60_−derivative as the ETL in PSCs without hysteresis effect, demonstrating the first inverted PSC with a PCE of 3.9% [30]. Although the PCE remains to be further improved, the inverted structure gives researchers some methods for optimizing devices. Therefore, extraordinary effort has been put in to enhance efficiency and property of inverted devices in the last few decades (Figure 1). In 2014, Bai et al. used PCBM/ZnO as the ETL with the optimal annealing process of perovskite layer, and the planar heterojunction MAPbI_3−x_Cl_x_ PSC exhibited PCE of 15.9%. In addition, the high PCE of 12.3% with a large−area (1 cm^2^) device was obtained [31]. In 2016, Chen et al. utilized the photoexcited carrier balance strategy through interface modification to improve the photoexcited charge transport, and a high PCE of 18.72% was achieved with the *N*,*N*−Dimethylformamide−treated (DMF−treated) PEDOT:PSS as HTL and poly(methyl methacrylate)−modified (PMMA−modifed) PCBM as ETL [32]. Afterwards, different approaches such as defect passivation, adjusting crystal size and interfacial engineering were intended to fabricate efficient inverted PSCs. In 2017, Zheng et al. treated ionic defects of organic–inorganic halide perovskite materials through quaternary ammonium anions and cations, which could effectively prevent the charge traps, prolong the life span of photoexcited carrier, as well as achieve a certified PCE of 20.59 ± 0.45% [33]. In 2018, Luo et al. reported that the nonradiative recombination of the inverted devices was effectively reduced by using a solution−processed secondary growth technique with bromide guanidine solution, and the maximum PCE of 21.5% with an V_oc_ of 1.21 V was eventually obtained [34]. In 2020, Li et al. reported that their inverted PSC showed the highest PCE of 23.37% (with 22.75% certified) by using the bifunctional molecule, piperazinium iodide (PI), to tailor the end groups on the surface of the absorbing layer and modify surface chemical properties [35]. Until now, Li et al. have fabricated the perovskite heterojunction by surface sulfidation [36]. Furthermore, the firm Pb−S bonds could effectively strengthen the perovskite heterojunction and the stability of the PSCs, which gave rise to an optimal efficiency of 24.3%, and the intensive stability with over 90% of the original efficiency under one thousand hours of illumination at 55 ± 5 °C.

Despite the rapid development of the inverted PSCs, there are a series of issues compared with the traditional upright PSCs, such as nonradiative recombination of carriers, stability issues, costly electron−transporting materials and hole−transporting materials. These issues should be fully solved in the following studies to obtain the excellent performance device. Thus, it is necessary to have a summary of the inverted PSCs. We discuss and summarize the latest research of inverted hybrid PSCs, as well as the all−inorganic inverted PSCs. Firstly, the architecture and working principle of inverted PSCs are presented. Then, the materials about inverted PSCs including charge transport materials and perovskite films are discussed. Since the properties of material change with the interface processing and material modification, we summarize their performance and progress in detail. In the end, the challenges of inverted PSCs and the strategies for improving performance are discussed.

## 2. Basics of Inverted Perovskite Solar Cells

### 2.1. Architecture of Inverted PSCs

Since PSCs originated from dye−sensitized solar cells (DSSCs), architecture and working principle for devices are similar to those of DSSCs. As shown in Figure 2, the architecture of devices could be categorized into the upright architectures (n−i−p) and the inverted architectures (p−i−n). Moreover, the upright architectures (n−i−p) could be further classified into mesoporous structures and planar structures. Architecture of inverted PSCs consists of transparent conducting oxide (TCO), HTL, an absorption film, ETL and the metal back electrode as counter electrode, which is similar to the architecture of traditional upright PSCs. Normally, the HTL contains conductive polymers and inorganic semiconductor materials. The materials for ETL show less selectivity, mainly the fullerene and fullerene derivatives. The selection of ETL and HTL is based on the accessibility of the alignment of energy structure and preparation process.

### 2.2. Working Principle of Inverted PSCs

The working principle of inverted PSCs can be expressed as follows [39,40,41]: (1) The perovskite absorber layer absorbs photons in light and creates electron–hole pairs under the excitation of light with enough energy, and then separates free carriers. The perovskite absorber layer, with a narrower bandgap, can absorb more sunlight and produce a higher photocurrent. However, this will give rise to lower open−circuit voltage due to the discrepancy of quasi−Fermi levels of carriers [39,40]. (2) The photoexcited holes are injected into the valence band (VB) of HTL and transferred to the TCO substrate due to the built−in electric fields, while the photoexcited electrons leap into the conduction band of ETL and transfer to the metal counter electrode for the reason given above. (3) The photoexcited electrons transfer to the cathode and the photoexcited holes are collected through the anode, which forms an electrical circuit. (4) However, the recombination process exists before hole injection to the HTL and electron injection to the ETL. As shown in Figure 3, before electron injection to the ETL, electrons in the absorber layer can recombine directly with holes in the absorption layer (r_1_) and with holes in the HTL (r_2_). In addition, the electrons in the ETL can react with holes in the absorption layer (r_3_), followed by recombination with holes in the HTL (r_4_).

## 3. Hole−Transport Layer (HTL)

In order to accelerate the extraction of photoexcited holes in the absorption layer and reduce the energy loss, the HTL is introduced on the surface of TCO substrate in the inverted PSCs. The ideal HTL must fulfill the following requirements [41,42,43]: (1) Matched energy level. The highest occupied molecular orbital (HOMO) levels of HTL should be close to the VB of perovskite absorber layer. (2) High hole mobility. The high hole mobility favors the efficient transportation of photoexcited holes from the absorption layer to TCO substrate. (3) Good transmittance. Considering that the sunlight should get through the HTL before being absorbed by perovskite film, reduction of light loss can maximize the absorption and utilization of light. (4) Good film uniformity. The uniform film is beneficial to the generation of the absorption layer, with excellent crystallinity on the HTL. (5) Solution processability. Solution methods should not only simplify the preparation processes, but also be suitable for the commercialization of inverted PSCs in the future. For inverted PSCs, the types of HTL can be divided into conductive−polymer HTL, organic small−molecule HTL and inorganic−semiconductor HTL according to the property of materials. Figure 4 illustrates energy levels of different HTLs in inverted PSCs, including conductive−polymer materials (e.g., polytriarylamine (PTAA)), organic small−molecule materials and inorganic p−type semiconductor materials (e.g., NiO_x_ and CuI) [44,45].

### 3.1. Conductive Polymers Materials

#### 3.1.1. Poly(3,4−Ethylenedioxythiophene):Poly(Styrenesulfonate) (PEDOT:PSS)

Conductive−polymer materials have turned out to be a type of important p−type semiconductor material and have been used in various categories of thin film photovoltaic devices, such as organic solar cells devices, DSSCs, PSCs and so on. PEDOT:PSS has been extensively explored as HTL for inverted devices because of its excellent electron conductivity, high transmittance, the matched energy level and low temperature annealing process. Thus, great effort has been made to study its application in inverted PSCs. Performances of several representative inverted PSCs on the basis of pure PEDOT:PSS are shown in Table 1 [30,37,46,47,48,49]. In 2013, PEDOT:PSS was first used as hole transport materials in the inverted PSCs by Jeng et al., the efficiency of 3.9% was obtained [30]. However, the PEDOT:PSS−based inverted PSC suffered a poor V_oc_ (0.88 ~ 0.95 V). In order to solve this issue, the pH value of PEDOT:PSS was controlled using imidazole [46,47,48]. Therefore, an improved efficiency of 15.7% with the V_oc_ of 1.06 V was acquired for Wang et al., and an efficiency of 14.25% with the V_oc_ of 0.994 V was acquired for Yi et al. Furthermore, stability of devices for long periods was enhanced due to the change in pH of PEDOT:PSS. Subsequently, Hu et al. prepared a PEDOT:PSS monolayers by using the water rinsing process on the ITO conductive substrate. A directional electric field from PEDOT with a positive charge to PSS with a negative charge was induced due to the Coulomb interaction of PEDOT and PSS, which can boost extraction of photoexcited holes. Furthermore, the PCE of 18.0% was yielded due to the enlarged V_oc_ and fill factor (FF) [49]. Heo et al. prepared the inverted PSC with the PCE of 18.1% by depositing compact pinhole−free MAPbI_3_ perovskite film on the HTL by the spin−coating process, which was the highest PCE as ever reported with regards to pure PEDOT:PSS−based inverted PSCs until now [37].

Although the pure PEDOT:PSS shows good potential application for photoelectric properties of the inverted PSCs, there are some disadvantages that need to be overcome. First, the Fermi level of pure PEDOT:PSS mismatches with the energy levels of the absorption layer, which leads to the inefficient extraction of photoexcited holes and the low V_oc_. In addition, the hygroscopic property and acidity of PEDOT:PSS film will affect PSCs for long periods of stability. Numerous strategies, including doping PEDOT:PSS [50,51,52,53,54,55,56,57,58,59,60,61,62,63,64], modified PEDOT:PSS [32,65,66,67,68,69,70,71,72,73,74], interface engineering [75,76,77,78,79,80] and composite hole transport materials [81,82,83,84,85,86,87,88] have been developed to solve these issues.

(1)Doping PEDOT:PSS

Doping of PEDOT:PSS has been widely employed to improve performance of photovoltaic devices. With regards to the inverted PSCs, the method of doping PEDOT:PSS was also studied to enhance properties of solar cells. Table 2 [50,51,52,53,54,55,56,57,58,59,60,61,62,64] shows the properties of several representative inverted PSCs on the basis of doping PEDOT:PSS with different substances. Doping could be applied to regulate the energy structure of HTL for inverted devices to enhance the hole transport performance of PEDOT:PSS and V_oc_ [50,51,52,53,54,55,56,57]. For example, in 2018, Tang et al. tuned the work function of PEDOT:PSS from −5.02 eV to −5.19 eV by doping the HTL with the perfluorinated ionomer (PFI), and the optimal PCE of 15.85% was obtained [53]. Liu et al. doped rubidium chloride (RbCl) into PEDOT:PSS to elevate the properties of HTL [54]. By controlling the content of RbCl, it could enhance the work function of HTL, electrical conductivity and hole transport capability, which could simultaneously decrease the phase disengagement of PEDOT:PSS and enlarge the grain size, leading to an increased efficiency of 18.3%. In 2019, Jiang et al. enhanced the separation of photoexcited charge and hole transportation through doping PEDOT:PSS with CsI [55]. As a result, the PCE beyond 20% and a better V_oc_ of 1.084 V were received for inverted PSC based on CsI−doped PEDOT:PSS. Meanwhile, doping PEDOT:PSS can improve morphology of the absorption layer [56,57,58]. For example, doping CuSCN into PEDOT:PSS not only led to the improvement of charge extraction efficiency, but also resulted in the rough surface topography of HTL. This induced the enlarged of perovskite crystalline sizes and the enhanced PCE [57].

Organic doping or p−type doping can also help to promote the electrical conductivity of PEDOT:PSS, including RuCl doped [54], Ag nano−particles doped [59], polyethylene oxide (PEO) doped [60], 2,3,5,6−tetrafluoro−7,7,8,8−tetracyanoquinodimethane (F4−TCNQ) doped [61], NaCl doped [62] and so on. In 2016, Hong et al. used the PEO−doped PEDOT:PSS as HTL to construct the inverted PSCs [60]. Doping CuSCN into PEDOT:PSS induced the improvement of its conductivity, which resulted in the effective transfer of photoexcited charge and the enhanced J_sc_. Based on the investigation by Liu et al., it is regarded that doping F4−TCNQ into PEDOT:PSS can effectively tune electrical performance and HOMO level of PEDOT:PSS film [61]. As a result, the composite film revealed the improved electrical conductivity and beneficial energy level alignment, and the increased photoelectric performance (including J_sc_, V_oc_, FF and PCE) was observed for inverted PSCs. Doping HTL with certain amount of NaCl could not merely contribute to enhancing the electrical conductivity of PEDOT:PSS, but help to induce the crystal orientation of perovskite layer along the (001) crystal plane on the film of HTL [62]. Recent studies suggest that the moisture stability and long−term stability can be enhanced through PEDOT:PSS [54,63,64]. In 2020, Xu et al. constructed the gradient heterojunction (GHJ) based on PEDOT:PSS/PEDOT:PSS−VO_x_ by doping the content of VO_x_ into PEDOT:PSS, resulting in the improvement of charge extraction and PCE [64]. Moreover, the GHJ−based PSC exhibited outstanding long periods of stability, which remained over 80% or 70% of the original efficiency under illumination in nitrogen atmosphere for 750 h or in the air for 175 h.

(2)Modified PEDOT:PSS

In addition to the strategy of doping PEDOT:PSS, modification has been proved to be beneficial to solve problems of the inferior V_oc_ and long−term stability. Performances of several representative inverted devices based on modified PEDOT:PSS are displayed in Table 3 [32,65,66,67,68,69,70,71,72,73,74]. Properties of PEDOT:PSS could be treated through solution processing [65,66,67,68]. PEDOT:PSS modified by dimethylformamide (DMF) solvent, Dimethyl sulfoxide (DMSO) solvent and the DMF solution of MAI was first reported by Xia et al., which led to the better electrical conductivity and superior surface of HTL [65]. As a result, it increased the J_sc_ while lowering FF and PCE of the inverted PSCs. Afterwards, researchers treated the PEDOT:PSS with these solutions by different approaches [32,66,67]. Chen et al. reported that PEDOT:PSS was rinsed with DMF to balance the carrier transport [32]. As a result, the charge transfer of devices was accelerated and balanced, which gave rise to the restraint of charge recombination at perovskite/selective contact interface and an improvement of PCE with 18.72%. Huang et al. modified PEDOT:PSS with certain percentage of DMSO, resulting in the PSC with a high light harvesting, and enhancing charge extraction and long periods of stability [67]. Apart from the common organic solvents, the use of 1−Ethyl−3−methylimidazolium chloride (EMIC) ionic liquids to promote the properties of PEDOT:PSS was reported firstly by Zhou et al. [68]. The treated PEDOT:PSS caused the excellent electrical conductivity, desirable surface morphology and lower work function for HTL, thereby obtaining the inverted PSCs with an increased efficiency.

On the other hand, performances of inverted devices could be promoted by modifying the chemical structure of HTL [69,70,71,72]. For example, in 2017, Huang et al. firstly applied dopamine (DA) to modify PEDOT:PSS to exploit a novel dopamine−copolymerized film [69]. As a consequence, it decreased the acidity of film, and improved the stability and PCE. Then, they further investigated the DA semiquinone radical treated PEDOT:PSS (DA−PEDOT:PSS) film and influence of DA doping on electron donating capability of DA−PEDOT:PSS [70]. The results of studies suggested that the DA−PEDOT:PSS exhibited enhanced charge extraction capability and work function, and the fabricated devices possessed improved V_oc_ and an impressive PCE of 18.5% with high stability. Hydrogen peroxide (H_2_O_2_) can also be used to oxidize PEDOT:PSS monolayer to elevate the charge transmission performance for absorption layer to electrode and limit in−plane change transport [71]. As a consequence, the inverted devices based on the oxidized PEDOT:PSS monolayer yielded an enhanced FF of 0.82 and a PCE of 18.8%. In addition, hydroxymethyl (MeOH) can be added as a functional group to modify ethylenedioxythiophene (EDOT) [72]. Therefore, the work function of modified HTL was improved, and the energy level alignment, electrical conductivity and surface topography of PEDOT−MeOH:PSS were enhanced. Recent reports have discovered that the properties of PEDOT:PSS could be modified by using urea and sodium benzenesulfonate (SBS) to tune its morphology and work function [73,74], resulting in an excellent surface morphology and suitable energy level arrangement with the absorption layer, and better crystallinity of the absorption film.

(3)Interface engineering

Apart from the modification, intercalation of interlayer film into the HTL can enhance the performance of inverted PSCs by interface engineering. Table 4 [75,76,77,78,79,80] shows the performances of several representative inverted PSCs on the basis of interfacial modification of the interlayer. First of all, the perovskite film will be in contact with the interlayer directly after the interlayer deposits on the PEDOT:PSS film, this will efficiently inhibit electron leak to restrain the charge recombination because of the high level of the lowest unoccupied molecular orbital (LUMO) about the interlayer. Jhuo et al. have employed the cross−linked organics *N*,*N*′−BIS(4−(6−((3−ethyloxetan−3−y)methox)−hexyloxy)peny−*N*,*N*′−bis(4−methoxyphenyl)biphenyl−4,4′−diamin (QUPD) and *N*,*N*′−biS(4−(6−((3−ethyloxetan−3−y)methoxy))−hexylpenyl)−*N*,*N*′−diphenyl−4,4′−diamn (OTPD) interlayers as electron blocking layers at the interface of HTL and absorption [75]. The HOMO level of cross−linked interlayers does match with that of the absorption layer, and the LUMO energy level of cross−linked interlayer was higher than that of HTL, which is beneficial to the hole collection and electron blocking. Second, the application of the interface between HTL and absorption layer, which can tune the energy structure mismatch, such as polyethyleneglycol (PEG) interlayer and *N*,*N*′−Bis−(1−naphthalenyl)−*N*,*N*′−bis−phenyl−(1,1′−biphenyl)−4,4′−diamine (NPB) interlayer [76,77,78], and FF can be improved [77]. Third, the inserting of the interlayer can strengthen the contact interface between PEDOT:PSS and absorption film, and then modify morphology of perovskite film [78,79]. Gu et al. applied 3−aminopropanoic acid as a monolayer with self−assembly (C3−SAM) on HTL [79]. As a consequence, C3−SAM could remarkably enhance crystallinity and coverage of the absorption layer. At last, interlayer at the HTL/absorption layer interface can inhibit the corrosion of perovskite precursors on substrates and the decomposition of absorption film due to the hygroscopic property and acidity of HTL, which resulted in an improvement of the stability for solar cells [80]. Luo et al. fabricated a graphene−oxide (GO) coating layer on the HTL [80]. The GO interlayer has been effectively enhanced the morphology of absorption layer and restrained decomposition of absorption layer. As a consequence, the PSC showed a PCE of 15.34% and the stability was obviously enhanced with efficiency remaining at 83.5% of the original value when exposed to the air up to 39 days.

(4)Composite HTL materials

The photoelectric properties of PEDOT:PSS−based inverted devices could also be strengthened with modifying another p−type hole transport material. Through this process, the other hole transport material can make up the shortcoming of PEDOT:PSS layer and contribute to improvement performance of devices: (1) Adjust the energy level arrangement in the architecture. (2) Promote the film quality of perovskite layer. (3) Accelerate the carriers transport at the HTL/absorption layer. (4) Advance the stability of devices. (5) Enhance the electron−blocking capacity. Performances of several representative inverted devices on the basis of the composite film are shown in Table 5 [81,82,83,84,85,86,87,88]. For example, V_2_O_5_ was used as the HTL to tune the energy level alignment in the inverted PSCs, which led to the increase of the work function (from 5.1 to 5.4 eV) and a boosted efficiency of 17.5% [81]. Nickel phthalocyanine (NiPcS_4_) was incorporated into HTL, resulting in the increase of perovskite crystallinity, charge transfer at HTL/absorption layer interface, and the stability in the air [82]. These factors led to efficiency up to 18.9%. Yoon et al. fabricated a hybrid HTL that consists of the single−walled carbon nanotubes (CNTs) and PEDOT:PSS for inverted PSCs [83]. The hybrid HTL exhibited a superior quality film and the enhanced electron−blocking properties, which increased the PCE from 13.2% to 16.0%. Li et al. developed a hybrid HTL by incorporating the oxidized carbon nanorods (OCNRs) into the HTL [84]. The hybrid HTL could tune energy level alignment between HTL and absorption layer, resulting in an enhanced efficiency of 19.02%. It is worth noting that most of the composite hole transport materials are based on PEDOT:PSS for the inverted PSCs so far.

#### 3.1.2. Poly(Bis(4−Phenyl)(2,4,6−Trimethylphenyl)Amine) (PTAA)

PTAA is regarded as another typical of conductive polymers HTM. These inverted PSCs which recently reported the highest efficiency are on the basis of PTAA. Table 6 [34,89,90,91,92,93,94,95,96,97,98,99,100,101,102] shows the performances of several representative inverted PSCs devices on the basis of the different PTAA HTL structures. On one hand, PTAA is the typical π−conjugated organic polymer material and its non−planar molecular structure is amorphous. This will lead to the dense and uniform film and excellent isotropic hole transport performance. On the other hand, the inferior V_oc_ of inverted PSCs is mainly because of nonradiative recombination. Furthermore there is no microstructural ordering during annealing for PTAA. This will reduce the nonradiative recombination in the interface. In addition, hydrophobic property of PTAA would favor the stability of the devices. However, some articles showed that the UV light can cause the PTAA degradation, which can interfere with the charge transport, as well as increased defect population upon prolonged UV exposure. So, some actions should be taken to improve its resilience, such as doping materials or adding a layer [103].

As early as 2015, Bi et al. systematically researched hydrophobic property of wetting and non−wetting polymer HTLs, such as polyvinyl alcohol (PVA), PEDOT:PSS and PTAA, the growth mechanism of MAPbI_3_ perovskite crystals on the HTL substrates [89]. The results indicated that MAPbI_3_ crystals on PTAA have significantly larger grains, higher crystallinity and fewer grain boundaries compared with that on the wetting HTL substrates. This will reduce the crystal defects and the nonradiative recombination in the absorption layer, resulting in the outstanding efficiency of 18.1% for PSC on the basis of the PTAA. Subsequently, in 2017, Serpetzoglou et al. targeted researched the hydrophilic PEDOT:PSS and the hydrophobic PTAA polymer HTLs, along with the corresponding inverted PSCs. The device on the basis of PTAA exhibited the better photoelectric performance with the PCE of 15.67%, but the device on the basis of PEDOT:PSS obtained the efficiency of 12.6% [90]. This is mainly due to the decrease of HTL surface roughness, and the hydrophobic performance of HTLs only explained a little about the difference in the photoelectric performance. By 2018, Luo et al. groundbreaking had reported that the nonradiative recombination of the inverted devices was effectively reduced by using a solution−processed secondary growth technique with bromide guanidine solution, leading to the highest PCE of 21.5% ever up to that time for PTAA/perovskite/PCBM/C_60_/BCP/Cu device [34].

However, PTAA as a kind of polymer semiconductor is now faced with the problem of poor electrical conductivity. Doping is an efficiently strategy to enhance properties of PTAA. Wang et al. reduced device series resistance by doping F4−TCNQ in PTAA, resulting in an improvement of FF and V_oc_ without sacrificing short circuit current [91]. However, the F4−TCNQ had the problems of poor solubility and high cost. Recently, Liu et al. have introduced a NPB film into PTAA to strengthen the wettability of PTAA and promote the surface topography of the absorption layer [92]. Consequently, the optimal PSC exhibited excellent photoelectric performance and obtained a PCE of 20.15%.

The poor electrical conductivity of PTAA exists not only at the ITO/PTAA interface, but also between the PTAA and perovskite film. Therefore, modification is also an efficient strategy to promote the performance of PTAA [93,94,95,96,97,98]. Reduced graphene oxide (rGO) is also widely doped into HTL by researchers [104]. Zhou et al. employed the stability of reduced graphene oxide (rGO) to construct the ITO/r−GO/PTAA hole−transport bilayer, leading to a decrease of defects in the absorption layer and an increase of PCE with 17.2% for the rigid inverted PSCs [94]. It is well−known that the monomolecular HTL often exhibits superior surface and electronic properties compared with the regular HTL. However, the study showed that the work function and conductivity of monomolecular HTL were decreased [97]. To solve this problem, Chen et al. introduced the 2,3,5,6−tetrafluoro−7,7,8,8−tetracyanoquinodimethane (F4−TCNQ) as the interlayer between PTAA and perovskite film, resulting to an improvement performance of device [97]. Polyelectrolyte is often used as the typical material for interfacial modification to enhance performance. Jung et al. used a series of anionic conjugated polyelectrolytes (CPEs) as the interlayer between PTAA and absorption layer [98]. As a result, the wetting and morphology of the absorption layer were enhanced because of the improved wettability of perovskite on the PTAA, so a large−area solar cell obtained a stable PCE of 18.38%. 

In addition, one of the most frequently used methods is interface engineering for improving the properties of HTL, especially for the surface properties of HTL [99,100,101,102]. Though the hydrophobic property of PTAA can promote the stability of the inverted devices, PTAA makes the precursor of absorption layer difficult to cover on the PTAA substrate due to its large surface energy, and many researchers have used the means of interface engineering to improve the surface activity of PTAA toward perovskites [99,100,101,102]. In 2018, oxygen plasma processing is employed to modify the hydrophobicity of HTL by Zhang et al. [99]. The oxygen plasma treatment not merely decreased the surface energy and enhanced work function of HTL but increased electrical conductivity of HTL. As a result of the treatment, the efficiency of charge extraction was enhanced, and the inverted PSCs with oxygen plasma treated PTAA acquired a PCE of 19.0%. The hydrophobicity of PTAA can also be improved via solvent treatment process. In 2020, Li et al. modified the surface of PTAA by toluene during solution processing, resulting in the enhancement of wettability of PTAA and the improvement of morphology [100]. Apart from that, it is conducive to the holes extracted from absorption film to HTL, and the ohm contact between the PTAA and perovskite film has been enhanced. As a consequence, inverted PSCs illustrated an excellent efficiency of 19.13%. Recently, Xu et al. have textured the PTAA by demixing of the mixed polymer solution with PTAA and polystyrene (PS), and obtained a textured PTAA/perovskite interface [102]. By adopting an antireflection coating on ITO substrate, the device based on the textured PTAA exhibited a superior PCE with 21.6%.

#### 3.1.3. Conjugated Polyelectrolyte (CPE)

As the most commonly used material, PEDOT:PSS has unavoidable shortcomings, such as acidity and hygroscopicity. This leads to the poor stability and repeatability for the PSCs. However, CPE with pH−neutral and preparation at low temperature, as a new kind of HTL, avoiding the disadvantages of PEDOT:PSS successfully. More importantly, the energy level of CPE could be adjusted, which is beneficial to reduce interface contact resistance at active layer/electrode interface. In addition, functional group in the side chains of CPE can ensure good solubility in water and alcohol. Table 7 [105,106,107,108] shows the performances of several representative inverted PSCs on the basis of the different CPE HTL structures. In 2015, Choi et al. employed CPE with the decoration of functional group as the HTL to prepare a CPE−K film, which was used in PSCs for the first time [105]. Their devices with the CPE−K resulted in a maximum efficiency of 12.51% with elevated stability. CPE with the decoration of different inorganic ions has an effect on the self−doping capacity, conductivity and the adjustment of work function for HTL. Thus, many CPE based HTLs with the decoration of inorganic alkali metal ions [106,109,110], such as SO_3_^−^K^+^, SO_3_^−^Na^+^ and COONa^+^, have been developed since the first report. By contrast, CPE with organic cations has been applied in PSCs by researchers [107]. The results demonstrated that CPE with organic cations shows an excellent wettability to the absorption film, leading to an excellent efficiency with 19.76% [106]. After that, in 2020, Zhang et al. incorporated K^+^ into the CPE (TB(K)) to enhance the defect−passivation properties [108]. The result is that devices based on TB(K) displayed a superior photoelectric property with an efficiency of 20.1% due to the favorable inhibition of defect state and collection efficiency of photoexcited holes for TB(K).

#### 3.1.4. Polyelectrolyte

The work function of PEDOT:PSS mismatches with energy levels of absorption layer according to previous reports, which leads to the energy loss at HTL/absorption layer interface and lower V_oc_ of solar cells [49,50,51,52,53,54,55,56]. In order to overcome this shortcoming, polyelectrolyte was developed as a substitute of PEDOT:PSS considering its excellent performance in the fields of organic light emitting diodes (OLEDs) and organic solar cells (OSCs). The introduction of polyelectrolyte can lead to the forming of dipole at the ITO/polyelectrolyte interface and regulate work function of ITO, resulting in an improvement of carrier extraction and photoelectric properties. Table 8 [111,112,113,114] shows the performances of several representative inverted PSCs devices on the basis of the polyelectrolyte structures. In 2015, Li et al. used water soluble polyelectrolytes (P3CT−Na) as HTL in PSCs to facilitate the performance for the first time, which resulted in an excellent performance with PCE of 16.6% due to the desirable match of energy structure and favorable crystal of perovskite film [111]. To modify surface wettability of P3CT−K, doping of graphdiyne in P3CT−K was carried by Jiu et al. in PSCs, resulting in an improvement of hole collection efficiency and a decrease of charge recombination [113]. Therefore, Jiu et al. further employed the small molecule ethanediamine to construct a novel polyelectrolyte (P3CT−ED). As a result, application of P3CT−ED can significantly improve the hole transport of P3CT−ED and the crystallinity of perovskite film, as well as reduce surface defects.

#### 3.1.5. Poly[*N*,*N*′−Bis(4−Butylphenyl)−*N*,*N*′−Bis(Phenyl)Benzidine] (Poly−TPD)

As the typical non−wetting hole transport materials, Poly−TPD possesses the higher LUMO energy level in comparison to PEDOT:PSS material, which give rise to an efficient collection of photoexcited holes and separation efficiency of photoexcited charge, as well as facilitating the crystal growth of perovskite materials. Thus, the large−sized perovskite crystalline and few lattice defects of perovskite films were obtained for PSCs based on Poly−TPD HTL by Zhao et al. [115]. Table 9 [115,116,117,118,119] shows the performances of several representative inverted PSCs devices based on the Poly−TPD HTL structures. Xu et al. adopted ultraviolet−ozone modification method to tune the surface wettability of Poly−TPD to improve the absorption layer with desirable crystallite dimension and favorable coverage, and a highest PCE of 18.19% was acquired [116]. Interface modification is an effective method of improving the interface contact and reducing the defect states between the HTL and the perovskite films [117,119]. You et al. introduced an insulating layer of Al_2_O_3_ nanoparticles to decrease the surface energy of hole transport materials, which resulted in perfect optoelectronic properties of HTL and the full V_oc_ [117]. Li et al. used the conjugated polyelectrolyte (PFN−I) to enhance the interface contact and reduce the defects at HTL/absorption layer interface and perovskite/ETL interface simultaneously, leading to the enhancement photoelectric properties and stability of PSCs devices [119]. An outstanding efficiency of 20.47% and over 80% of the original efficiency were observed for the prepared devices with 800 h under humid conditions, with 35–55% at room temperature.

#### 3.1.6. Other Conductive Polymer Materials

High cost is a problem which cannot be ignored for the most popular conductive polymers HTL. For example, PEDOT:PSS and PTAA. With that in mind, other conductive polymers are also used as the HTL of inverted PSCs, such as polythiophene and poly(p−phenylene) (PPP) [120,121]. Table 10 [120,121,122,123,124] shows the performance of several representative inverted PSCs on the basis of other conductive polymers HTL. Yan et al. deposited the prepared the polythiophene on the ITO surface as the HTL via electrochemical polymerization, and the inverted PSC based on polythiophene showed a promising efficiency of 15.4% [120]. They further prepared a sequence of conductive polymers, such as PPP and polythiophene (PT), as the HTL using electrochemical polymerization [121]. The constructed devices based on PPP exhibited an outstanding V_oc_ with 1.05 V and an efficiency of 16.5% due to its work function of −5.31 eV. Wang et al. adopted conjugated polymer poly(9−vinylcarbazole) (PVK) as the HTL, and found that PVK based devices had better stability in contrast to PEDOT:PSS based devices [122]. In addition, a superior charge recombination resistance of PSCs was obtained due to the fact that perovskite film has preferable crystal properties and fewer PbI_2_ surplus. Liu et al. prepared two new types of nonlinear π−conjugation molecules as the HTL, which were Y−shaped (XSln847) and X−shaped (XSln1453) [123]. Compared to the X−shaped molecule, a Y−shaped molecule has superior charge transfer and reduced charge recombination due to the fact that XSln847 molecule forms a compact stacking arrangement of molecules through short contacts between molecules to achieve a nest−layer in molecules. As a consequence, an efficiency of 17.16% was acquired with XSln847 based PSC.

In addition to them, the method of dopants is considered to be an effective strategy to settle the defects of hole mobility and charge recombination for other conductive polymers materials. Therefore, Shao et al. developed a low−cost in situ electropolymerized polyamines (poly−1) as HTL for inverted PSCs [124]. The results of research suggested that the undoped poly−1 has superior hole collection ability and carriers transport properties, as well as strong hydrophobicity. As a result, the inverted PSCs with the optimal undoped poly−1 exhibit excellent long−term stability and a greatest efficiency of 16.5% compared with that for undoped PTAA.

### 3.2. Organic Small−Molecule Materials

Compared to the conductive polymer materials with high cost and complex synthesis process, small−molecule hole−transport materials not only have the advantages of simple synthesis, low cost and easy purification, but also can optimize the chemical structure and photoelectric performance by molecular engineering, which has gradually attracted the interest of researchers [125]. With the efforts of researchers, the efficiency of inverted PSCs based on organic small−molecule materials is increasing, and it tends towards catching up with PTAA. Table 11 [126,127,128,129,130,131,132,133,134,135,136,137,138] shows the performances of several representative inverted PSCs based on organic small−molecule HTLs. Instead of PEDOT:PSS, Li et al. employed two kinds of original micro−molecule organics 4,4′−bis(4−(di−p−toyl)aminostyryl)biphenyl (TPASBP) and 1,4′−bis(4−(di−p−toyl)aminostyryl)benzene (TPASB) as the HTL in inverted devices [126]. In comparison with PEDOT:PSS, TPASBP and TPASB with linear π−conjugated structure showed efficient hole transport properties, and the perovskite film on them presented large crystals and reduced grain boundaries. Hence, an efficiency of 17.4% and 17.6% was obtained for inverted devices based on TPASBP and TPASB HTLs. Yang et al. used the micro−molecule organics 4,4′−cyclohexylidenebis[*N*,*N*−bis(4−methylphenyl) benzenamine] (TAPC) as HTL in inverted PSCs [127]. Smooth, uniform, and hydrophobic TAPC with π−conjugated structure were achieved with the optimization of solution concentration and annealing temperature, leading to an outstanding efficiency of 18.8%. Zhang et al. prepared and adopted four diphenylamine derivatives with a fluorene core as HTL in inverted PSCs [128]. The insoluble 3D networks can be formed for the synthesized HTL with vinyl crosslinking via a suitable annealing temperature, leading to a low solvent resistance for preparation and a better stability. Hence, the device with the optimized HTL illustrated the prominent efficiency of 18.7% with the highest V_oc_ of 1.15 V. Li et al. used the micro−molecule organics (NPB) without dopants as HTL in MAPbI_3_ based inverted devices, leading to a prominent absorption layer with better crystallization properties, as well as a best PCE of 19.96% [129]. In 2019, Cao et al. showed an organic micro−molecule with spiro−based dopant−free (DFH) as HTL, which has a suitably positioned HOMO level for effective charge transfer from perovskite and the functional groups to adjust the conductivity and hole mobility [130]. As a result, the charge recombination was inhibited and an outstanding efficiency of 20.6% was achieved. In addition, DFH has an obvious advantage in price compared with other commonly used materials. Soon after, Wang et al. succeeded in increasing the efficiency of the inverted PSCs based small−molecule HTL to over 21% [131]. Not only the reported MPA−BTI and MPA−BTTI molecules have excellent photoelectric properties, but also the Lewis base groups in the molecules could inhibit defect states of absorption layer, which will give rise to enhanced performance of devices. With development of all−organic PSCs, Jiang et al. fabricated a novel 3D micro−molecule organics (TPE−S) as HTL in inverted all−organic CsPbI_2_Br PSCs, which exhibited a greatest efficiency of 15.4% because of the enhanced photoelectric properties, superior interfacial energetics, matched energy band structures and reduced charge recombination [132]. Furthermore the inverted hybrid PSCs obtained up to a PCE of 21.0%. Therefore, the reasonable design of molecular structure can achieve effective preparation for inverted devices on the basis of small molecules.

### 3.3. Inorganic Semiconductor Materials

#### 3.3.1. NiO_x_

NiO_x_ is a typical P−type inorganic material with a wide bandgap, usually representing a mixture of nickel (IV) and nickel (III) oxide (NiO_x_), which is the focus of current researches among P−type inorganic semiconductors. The researches show that NiO_x_ has high transmissivity, efficient charge extraction capability, and the matched energy level with perovskite [139,140,141]. Table 12 [140,141,142,143,144,145] shows the performances of several representative inverted PSCs based on NiO_x_. Docampo et al. adopted NiO_x_ as HTL in the inverted PSCs through calcination of nickel acetate tetrahydrate and monoethanolamine on a conductive substrate for the first time [139]. However, the efficiency of constructed device is less than 1%. This can be attributed to that the NiO_x_ in inverted PSCs is difficult to support an enough thick perovskite absorption layer, leading to a relatively inferior covering surface of absorption layer on NiO_x_ substrate. Thus, the enhancement of anchoring capability for NiO_x_ is the key to improving the NiO_x_ based devices. For this matter, Zhu et al. synthesized the NiO_x_ layer with high transparency as HTL via a sol−gel process [140]. The prepared NiO_x_ nanocrystals have a corrugated surface and well contacted with FTO substrates, which could support a sufficiently thick perovskite (300 nm) film. As a result, an excellent efficiency of 9.11% was acquired for the inverted devices on the basis of NiO_x_ nanocrystals HTL. However, the photoelectric performances of inverted PSCs on the basis of the NiO_x_ are still far behind that of inverted PSCs based on polymers materials and PSCs based on n−i−p structure. For this reason, in 2015, Park et al. improved the efficiency to as high as 17.3% with the FF value of 0.813 by preparing a well−ordered nanostructured NiO_x_ with a pulsed laser deposition method, which was the highest efficiency achieved at that time for the inverted PSCs based on P−type inorganic materials [141]. However, the preparation method with pulsed laser deposition method is inapplicable to the fabrication of large−scale devices. Therefore, Yin et al. adopted a small nanocrystal NiO_x_ film with cubic structure as HTL via a solution process with optimized concentration [142]. The obtained NiO_x_ film, with high conduction band, exhibited a superior electron blocking property, leading to a cell with less hysteresis, better air storage stability and 14.42% PCE. Islam et al. employed the sputter−deposited method to prepare the NiO_x_ film as HTL [143]. With optimized preparation condition of NiO_x_, the inverted devices based on the polycrystalline NiO_x_ film showed a better stability and a high PCE of 15.2%. Tang et al. prepared the NiO_x_ film with superior transmittance via sol−gel method, and its matched energy level and excellent optical transmittance led to a high PCE of 18.15% for the inverted devices [144]. Mali et al. prepared P−type nanoporous NiO_x_ (np−NiO_x_) film via a coprecipitation process, and an excellent efficiency of 19.1% was acquired for the inverted devices on the basis of np−NiO_x_ [145]. More importantly, more than 80% initial efficiency was restrained with 160 days at the air environment. Therefore, the preparation method played an important part in the properties of nickel oxide and photovoltaic devices, application of appropriate methods and optimization of preparation conditions can fabricate photovoltaic devices with excellent efficiency.

(1)Doping NiO_x_

Element doping is one of the most efficient ways to improve photoelectric performance of NiO_x_ HTLs. Doping elements are usually alkali metals (e.g., Li^+^, K^+^, Cs^4+^), alkaline earth metals (e.g., Mg^2+^), transition metals (e.g., Y^3+^, Cu^2+^, Ag^+^) and micro−molecule organics (F6TCNNQ), as is shown in Table 13 [146,147,148,149,150,151,152]. The advantages of doped NiO_x_ film are as follows: (1) Higher conductivity can effectively accelerate the extraction and transfer of photoexcited charges at NiO_x_/perovskite interface and reduce the series resistance of devices; (2) The work function of NiO_x_ film is closer to top of the VB of perovskite film, resulting to the higher hole mobility; (3) The transmittance of NiO_x_ film is improved, and absorption layer on NiO_x_ layer has better crystallinity and higher coverage.

Copper (Cu) was incorporated into the NiO_x_ film as a dopant by Kim et al. [146]. Doping of copper ions could efficiently promote the electroconductivity of NiO_x_, which accelerates extraction of photoexcited holes at NiO_x_/perovskite interface and reduces the series resistance of PSCs. It turned out that the efficiency of copper (Cu)−doped NiO_x_ (Cu:NiO_x_) based device was increased to 15.4%, as well as the J_sc_ and FF. In 2015, Chen et al. added lithium (Li^+^) and magnesium (Mg^2+^) to conduct heavily doped the NiO_x_ film to promote photoelectric performance of NiO_x_. Strategy of lithium−magnesium (Li^+^ − Mg^2+^) doping NiO_x_ film (Li_0.05_Mg_0.15_Ni_0.8_O) enhanced the extraction of photoexcited charge, and avoid the morphology of pinholes, as well as the local structural defects of PSCs for the large area devices [147]. In addition, Mg^2+^ doping compensates the band shift caused by Li^+^ doping into the lattice, leading to a high PCE of 18.3% and a large−area (˃1 cm^2^) device with 16.2%. Cesium (Cs^2+^) doped NiO_x_ film (Cs:NiO_x_) was conducted to prepare the HTL in inverted PSCs devices by them in 2017 [148]. The introduction of Cs^2+^ ions improved the conductivity and work function of NiO_x_ film, and finally realized a high PCE of 19.35%. In 2018, Wei et al. reported a novel inorganic HTL based on Ag^+^ doped NiO_x_ (Ag:NiO_x_) [149]. Transmittance, work function, electroconductivity and hole mobility of NiOx film could be elevated by replacing Ni site with Ag^+^ (Ag_Ni_) and behaving as the acceptor in NiO_x_ lattice. Absorption layer on Ag:NiO_x_ HTL show superior crystals, preferable covering surface and smoother topography, leading to a PCE of 16.4%, consequently. Yttrium (Y) was incorporated into the NiO_x_ film as a dopant via sol−gel method by Hu et al. [150]. The obtained Y:NiO_x_ HTL exhibited high hole mobility and quality morphology of perovskite film, which promoted interface charge recombination and transfer, and achieved an efficiency of 16.31%. Chen et al. successfully introduced an organic small molecule (F6TCNNQ) into the preparation of NiO_x_ film, which improved Fermi energy level of NiO_x_ and reduced energy level offset between NiO_x_ and absorption layer [151]. Consequently, the efficiency of CsFAMA based devices was raised to as high as 20.86%.

(2)Interface modification

Interface modification is another efficient way to promote photoelectric properties of NiO_x_, which can avoid possible uneven doping and unmanageable disorders caused by element doping, as is shown in Table 14 [153,154,155,156,157]. A few years earlier, interfacial layer was employed to promote the deficiency of NiO_x_ film in the inverted devices [153,154]. In 2015, Chen et al. prepared an ultra−thin NiO_x_ film through spray pyrolysis method and introduced an inert mesoporous alumina (Al_2_O_3_) with high transmittance as the barrier layer, leading to a higher PCE of 13.5% [153]. However, the obtained photoelectric property of inverted devices is far behind that for other structures HTL. Considering the problem of serious interfacial recombination, Chen et al. employed a potassium chloride (KCl) interface modification to reduce the interfacial recombination [155]. The modification of KCl led to an enhancement of absorption film and decrease of defect/trap density, resulting in an obvious enhancement in the V_oc_ from 1.07 eV to 1.15 eV. Lian et al. used an insulating film of PTAA film to improve the contact at NiO_x_/absorption film interface to decrease defects, which led to a better energy level arrangement, a deeper HOMO of NiO_x_/PTAA, less interface energy loss and a superior V_oc_ as high as 1.19 eV [156]. In 2020, Ru et al. employed the contrivable molecules (F2HCNQ) to adjust the conductivity and energy level of NiO_x_ film, resulting in a highest PCE of 22.13% with an FF of 82.8% [157]. Therefore, the designing of appropriate interface modification is an efficient method of acquiring excellent performance of inverted devices.

#### 3.3.2. Cu−Based Materials (CuX)

Cu−based materials (CuX), as typical P−type materials, have high carrier mobility and carrier diffusion length. More importantly, the matched energy level of CuX HTLs can effectively prevent the leakage of photoexcited electrons and reduce the energy loss of perovskite solar cells, especially for copper thiocyanate (CuSCN) and copper iodide (CuI). Hence, several representative inverted devices on the basis of the Cu−based HTL structures were reported, as is shown in Table 15 [158,159,160,161,162,163,164,165,166,167,168,169]. In 2014, Subbiah et al. showed the electrodeposited CuSCN as HTL in inverted devices firstly, resulting in the PCE of 3.8% [158]. However, the obtained PCE is far behind that for the organic HTLs based devices. Ye et al. fabricated a high−quality absorption layer on the electrodeposited CuSCN film, which led to a smaller surface roughness of absorption layer and lower interface contact insistence at CuSCN/absorption film interface [159]. It turned out that the average PCE was promoted to 15.6% with the best efficiency of 16.6%. CuI, whose energy level structure was similar to CuSCN, is also used as HTL in inverted PSCs [160,161,162,163]. Chen and Sun et al. used a spin coating CuI as HTLs in inverted devices, leading to the high efficiency of 13.58% and 16.8% [160,161]. Wang et al. exposed the heat evaporated Cu layer to iodine steam to prepare a uniform CuI film, and used the CuI as the HTL of inverted devices, achieving an efficiency of 14.7% in CuI/MAPbI_3_/PCBM/Au structure [162]. In 2017, Ye et al. first inserted a original p−type Cu(thiourea)I (Cu(Tu)I) as trap state passivators (TSPs) into the absorption film for the inverted PSCs to decrease the trap states of absorption film, as is shown in Figure 5 [163]. The trap states of absorption layer were passivated through interaction of Cu(Tu)I and uncoordinated metal cations and halide anions on the surface of absorption film, and depletion width of p−i bulk heterojunctions was increased, which led to the acceleration of excited hole transfer, the reduction of charge recombination and a breakthrough certificated PCE of 19.9%.

P−type Cu_x_O has a narrower bandgap (2.2 eV) and the work function of −5.4 eV in comparison to CuSCN and CuI. The Cu_x_O crystal with small size is easy to get high quality film with uniform surface topography, which is conducive to the crystalline grow of perovskite film. Wu et el. prepared an ultrathin Cu_2_O thin films via thermal oxidation method and applied in the inverted devices with CuO_x_/MAPbI_3_/PCBM/Ag structure [164]. The photoelectric performance of devices is extremely vulnerable to thickness of Cu_2_O film, and an efficiency of 11% was acquired with the thickness of 5 nm Cu_2_O films through precisely optimizing the thickness of Cu_2_O layer due to the high hole mobility, matched energy structure with perovskite film and long lifespan of photoexcited charges. In 2017, Yu et al. prepared CuO_x_ film as HTLs in inverted PSCs via a solution process, and the high quality of perovskite film was achieved based the CuO_x_ film [165]. As a result, the hole extraction was improved and the stability of devices was enhanced, which lead to the high performance of device with a PCE of 17.43%. In addition to the above commonly used Cu−based materials, other Cu−based materials were also developed as HTLs in inverted devices, for instance CuS and Cu_3_PS_4_ [166,167]. Rao et al. firstly applied CuS nanoparticles as HTL in inverted PSCs using the solution processing, which can effectively improve the extraction of photoexcited holes and achieve a PCE of 16.2% with optimization [166]. Yin et al. firstly applied Cu_3_PS_4_ nano materials as a novel HTL for inverted devices, which was conducive to growth of absorption film on the Cu_3_PS_4_ HTL and leads to a maximum PCE of 18.17% with less hysteresis [167].

Compared with NiOx, Cu−based materials have a relatively narrow band gap (2~3 eV), which is beneficial to inhibit the recombination of photoexcited charge due to the reverse transport of photoexcited holes. However, the Cu−based materials have higher requirements on the quality, thickness and transparency of HTLs. Compared with other hole transport materials, V_oc_ and FF are relatively low, which can be ascribed to low charge collection at HTLs/absorption layer interface and large series resistance at the interface. Hence, interface modification, the hole transport bilayer and ion doping can be used to improve the photoelectric properties of HTL. In 2018, Wang et al. incorporated the CuSCN into the CuI HTL by a solution process method, resulting in high quality HTLs with higher conductivity and an impressive PCE of 18.76% [168]. Javaid et al. employed a hole transport bilayer composed of PTAA and CuI in the inverted PSCs devices, which gave rise to a high grade absorption layer with larger crystals and a highest efficiency of 20.34% [169].

#### 3.3.3. Other Inorganic Semiconductor Materials

In addition to NiO_x_ and Cu−based materials, other inorganic semiconductor materials were used as the HTL for inverted devices, for instance V_2_O_5_, MoO_x_ and CoO_x_, as is shown in Table 16 [170,171,172,173,174,175,176,177,178]. Graphene oxide (GO) with applicable work function (−4.9 eV) is used as a HTL of inverted PSCs, which can effectively improve the film quality of perovskite layer [170,171,172,173,174]. Wu et al. employed GO as a HTL in the inverted devices for the first time, resulting in high grade absorption layer and the PCE of 12.4% [170]. Yeo et al. demonstrated the reduced graphene oxide (RGO) as HTL through a solution and room−temperature process [171]. The inverted devices consisting of a structure of rGO/MAPbI_3_/PCBM/BCP/Ag exhibited an enhanced PCE of 10.8% with good reproducibility and desirable stability. Chen et al. employed oxo−functionalized (oxo−G_1_) as HTL in inverted devices by a solution process to solve the stability issues of PSCs [172]. As it turned out, oxo−G_1_−based inverted devices illustrated an obviously enhanced stability and an efficiency of 15.2% with the outstanding V_oc_ of 1.1 V. Wang et al. treated GO with ammonia via a general strategy to prepare the HTL for the inverted devices, considering the acidic nature of GO [173]. Consequently, the devices fabricated on the basis of ammonia−treated GO (a−GO) obtained enhanced PCE of 14.14% and environmental stability. In 2021, Csatriotta et al. employed potassium−doped graphene oxide (GO−K) as the interlayer between the mesoporous TiO_2_ and the perovskite layer and used infrared annealing (IRA). The final PCE was 18.30%, and the hysteresis of the devices was significant reduced [179].

V_2_O_5_ has a narrow band gap (2.0 eV) and a low work function (−5.2 eV), which could gain the effective hole extraction junction with absorption film and has been investigated to replace the frequently used HTL in inverted PSCs [139,174,175,176]. Docampo et al. used V_2_O_5_ as the hole transport materials for inverted devices with the construction of V_2_O_5_/MAPbI_3−x_Cl_x_/PCBM/TiO_x_/Al, and the PCE was less than 1% [139]. Guo et al. fabricated V_2_O_5_ thin films through a solution process method and obtained device with PCE of 5.1%. After that, they fabricated thin V_2_O_5_/PEDOT nanoribbons via a cocoon−to−silk−fiber reeling process as HTL for inverted devices, resulting in a high PCE of 8.4% [175]. Duan et al. prepared a hole transport bilayer composed of V_2_O_5_ and P3CT−K as HTL in the inverted PSCs, which gave rise to the boosting efficiency of 19.7% [176]. Furthermore, bilayer−based PSCs showed prominent stability, which restrained more than 86% of the initial efficiency without encapsulation in N_2_ for more than one month. Anizelli et al. reported that luminescent−down−shifting quantum dots enable filtering of UV radiation with increased solar cell stability [180]. The PCE of the non−encapsulated device with application of luminescent−down−shifting layer dropped by ~18% over 30 h, which was compared to ~97% for an unfiltered device, also without encapsulation. In addition, Guo et al. reported photoluminescent materials can be directly added to monitor the performance of PSCs [181]. They found that photoluminescent spectroscopy was a more sensitive method than UV visible light absorption for characterizing the initial stages of perovskite degradation. Mahon et al. designed the useful technique of tracing photoluminescence kinetics under continuous illumination at the “seconds−to−minutes” timescale, which was able to apply for analysis of PSCs at various steps of their fabrication and lifespan [182].

To avoid the issue of stability, the thermal evaporated MoO_x_ has been developed as HTL by Tseng et al. [177]. Furthermore, the surface coverage of perovskite film has been enhanced with the ultraviolet−ozone process, which led to a high PCE of 13.1%. Shalan et al. presented an ultrathin CoO_x_ film as the effective HTL in inverted PSCs by using the solution processing [178]. Introduction of a CoO_x_ layer has been used to effectively improve the perovskite layer with uniform and well−packed film and decrease charge recombination synchronously. More importantly, CoO_x_−based devices exhibited an enhanced efficiency of 14.5% and outstanding long−time stability, which retained a PCE about 12% for over 1000 h.

It is well−known that the HTL are essential interfacial contact layers between perovskite and electrode for inverted devices. Selective transport of holes by HTLs can not only accelerate the extraction of photoexcited holes at HTL/absorption layer interface and reduce the energy loss, but also efficiently inhibit electrons leak for photoexcited carriers. At present, an increasing number of effective and low−cost conductive polymer materials, such as polymethyl methacrylate (PMMA) and self−assembled monolayers (SAM), were developed as alternative HTL for inverted devices, which are wettable with perovskite materials or their precursors and beneficial to the growth of perovskite films with high quality. In addition, their softness makes them more compatible with flexible devices. Compared with conductive polymer materials, most of inorganic P−type semiconductors have low work functions, superior carrier mobility, excellent optical transparency and stability. The extraordinary advantages are conducive to the fabrication of high efficiency inverted PSCs with exceptional stability.

## 4. Perovskite Absorption Layer

Inverted PSCs and the traditional upright PSCs have the same structure and requirements for the perovskite absorption materials, so the inverted PSCs can adapt to the perovskite absorption materials that the traditional structure can adapt to. The perovskite absorption material of PSCs should satisfy the structure of ABX_3_, as Figure 6 shows. A cation is situated in the center of a cubic crystal, and 12 halide (X) ions around the A cation to form a coordination cubic octahedron. The B−site metal ion is located at the apex of the cubic crystal and is surrounded by six halide (X) ions as a coordination octahedron. The A ions are usually organic cations or metallic cations in PSCs, such as CH_3_NH_3_^+^ (MA^+^) and HC(CH_2_)_2_^+^ (FA^+^), Cs^+^ and Rb^+^. B ions are usually divalent metallic ions, including Pb^2+^ and Ge^2+^, etc. X ions are halogen ions, including Cl^−^, Br^−^ and I^−^. MAPbI_3_ is the most widely used perovskite material in the early stage. However, MAPbI_3_ decomposes easily in hot and humid environment, which affects the long−term stability of the device [183]. Instead, FAPbI_3_ has good resistance to high temperature and humidity, and a narrow band gap can expand the absorption region [184]. However, the issue is that the pure FAPbI_3_ phase is unstable in the cubic crystal structure and easily transforms into the hexagonal crystal structure, which greatly weakens its capacity of light absorption. Adding some ions with small radius, such as MA^+^, Cs^+^ or Rb^+^, to obtain other combinations, such as RbCsFA, RbMAFA, CsFAMA and RbCsMAFA, is an effective method to stabilize FAPbI_3_ phase and inhibit the phase transformation [157,185]. Alternatively, the all−inorganic perovskite film could be acquired by completely substituting organic cations with Cs^+^ ions, such as CsPbI_3_, CsPbI_2_Br, CsPbIBr_2_ and CsPbBrThe best B−site metal ion is Pb^2+^, but its toxicity is not suitable for commercial production in the future. Replacing the Pb^2+^ ion with other non−toxic metal ions is an effective method to solve this problem. However, Sn^2+^ is unstable and can easily be oxidized into Sn^4+^ in the air to form self−doping, which leads to the decomposition of absorption layer and the low efficiency of the PSCs [186]. If Ge^2+^ and Bi^2+^ are used to replace Pb^2+^, it will lead to the low efficiency due to the serious recombination of carriers. The radius of halide ions can increase the lattice constant of perovskite materials. The diffusion length of carriers can be increased from 100 nm to 1 μm after Cl^−^ doping in MAPbI_3_, and the formation of defects can be reduced [187]. The introduction of Br ions can increase the bandgap of perovskite materials, reduce the J_sc_ and increase the V_oc_ [188], and a series of perovskite films with different absorption bands could be obtained to regulate the light absorption range and carriers transport of perovskite materials by adjusting the proportion of halogen ions in perovskite materials.

Apart from composition of perovskite film, preparation and processing of perovskite films also play a decisive part in the properties of perovskite film. According to reports, defects on the surface or crystal boundary of perovskite crystals are caused by the solution process, which leads to non−radiative recombination of carriers and degradation of device efficiency [189]. The primary defect states include deep level defects, undercoordinated halide ions, undercoordinated Pb^2+^ ions, Pb−I anti−site, MA vacancies, halide ions vacancies and shallow level defects. Moreover, surface or grain boundary passivation is a common method to solve above defects of perovskite materials. In addition, various processes have been used to inhibit non−radiative recombination of carriers, such as the increase of grain size, ion compensation, secondary growth, heterojunction engineering and 2D/3D mixing, which are the most common methods [190,191,192]. Therefore, in addition to the superior photoelectric performance of perovskite materials, reasonable interface engineering and interface modification are also important ways to improve photoelectric performance of devices.

## 5. Electron−Transport Layer (ETL)

In inverted devices, separation and transfer of photoexcited electrons at perovskite/ETL interface play a decisive part in photoelectric properties of devices. In order to meet the development trend of large−area inverted and flexible devices, the ETL ought to conform following conditions: (1) Meet energy level matching of the perovskite/ETL and the ETL/metal electrode to promote the separation and transfer of photoexcited electrons. (2) High electron mobility to promote the fast transfer of electrons. (3) The deposition of ETL does not affect the properties of perovskite layer. Figure 7 shows the electron transport materials commonly used in inverted PSCs devices, including C_60_ and its derivatives as well as other non−fullerene electron transport materials.

### 5.1. PCBM

Fullerene (C_60_) and its derivatives are the most frequently applied ETL materials for inverted PSCs because of their applicable energy levels, excellent electron mobility and simple film forming process, as is shown in Table 17 [89,118,147,193,194,195,196,197,198,199,200,201,202,203,204]. In 2013, Jeng et al. used the C_60_ and C_60_−derivatives as ETL in the inverted devices firstly, demonstrating the first inverted PSCs based on PCBM with a PCE of 3.9% [30]. You et al. conducted low−temperature annealing treatment on PC_61_BM and improved the efficiency of inverted devices from 3.9% to 11.5% [193]. Chiang et al. prepared high grade perovskite layer by two−step solution processing in air atmosphere, and replaced PC_61_BM with PC_71_BM, making the PCE up to 16.3% [194]. However, PCBM has the inherent defects as the electron transport materials, for instance poor electrical conductivity and serious charge recombination. According to the reports, interface modification and doping modification are effective methods to solve the problems in PCBM.

Functions of interface modification are follows: (1) Enhancing the ohmic contact at ETL/metal film interface, promoting charge transport of photoexcited electrons. (2) Preventing the leakage of photoexcited holes. (3) Eliminating the local structural defects of PCBM layer to achieve the fast migration of carriers. In 2014, Wang et al. introduced an ultra−thin film of bathocuproine (BCP) at PCBM/metal electrode interface, which could not just improve direct contact between PCBM and metal film, but effectively prevent recombination of holes and electrons at the interface [195]. However, the researches confirmed that compared with BCP, the contact between C_60_ and PCBM had a more significant improvement in photoelectric properties. Cui et al. inserted a film of C_60_ between PCBM and Al. A PCE of 16.6% was obtained with the device of P3CT−Na/MAPbI_3_/PCBM/C_60_/Al [196]. On the basis of the above researches, Bi et al. obtained the high efficiency of 18.1% by inserting C_60_ and BCP into the PCBM/Ag [89]. Lee et al. deposited a film of LiF at PCBM/Al interface to promote the electron transfer and facilitate the ohmic contact at the ETL/metal electrode interface [118]. Furthermore, the PCE of 17.2% was obtained with the structure of Poly−TPD/MAPbI_3_/PCBM/LiF/Al. Chen et al. achieved the rapid migration of carriers and eliminated the local structural defects of the PCBM by inserting a charge extraction layer of Ti(Nb)O_x_ between PCBM and Al, achieving a PCE of inverted devices up to 18.3% and retaining over ninety percent of initial efficiency under 1000 h of illumination [147]. Ren et al. employed a layer of π−extended phosphoniumfluorene electrolytes (π−PFEs) as hole barrier layer between PCBM and metal electrode, which introduced a dipole momentum between PCBM and metal electrode, and strengthened the built−in field of PSCs [197].

In addition to the interface modification, doping modification, such as n−type dopant and surfactant dopant, is another method to promote the electrical conductivity and electron mobility of electron transport materials, strengthen surface coverage of PCBM on the absorption layer and reduce the charge recombination. Ji et al. used the graphene−doped PCBM as ETL for inverted devices improve the transmission performance of photoexcited electrons for the first time, pursuing the PCE of 14.8% [198]. Afterwards a surfactant, oleamide, was doped into the PCBM to strengthen the surface coverage of PCBM on the absorption film and interfacial contact between absorption film and metal electrode by Xia et al., making an efficiency of inverted devices up to 12.69% [199]. Bin et al. achieved the n−type dopants with low concentration of 13−dimethyl−2−phenyl−2,3−dihydro−1h−benzimidazole (N−BMDI), revealing the enhancement of its conductivity, the filling factor, the V_oc_ and a highest PCE of 18.1% [200]. Chen et al. improved the conductivity and reduced the intensity of photoluminescence by doping PCBM with CoSe, thus enhancing the ability of electron extraction [201]. Kakavelakis et al. doped the PCBM with rGO, enhancing the electrical conductivity of the PCBM fivefold. Moreover, rGO passivated the surface traps of perovskite, which resulted in the reduction of the light−soaking effect by a factor of three. Finally, they obtained the PCE of 14.5%, which was the highest PCE for the hysteresis devices [202]. Wang et al. used the PC_61_BM:SnO_2_ as ETL in inverted devices [203]. As a result, an advancement of electron transmission and a decrease of charge recombination were revealed due to the fact that the surface morphology of PCBM film was improved and the deep level defects of SnO_2_ were inhibited, which led to an outstanding efficiency of 19.7%. Yang et al. doped PCBM with n−type polymer material (F8TBT) to achieve a good coverage of PCBM on the perovskite film, achieving an efficiency up to 20.6% [204]. In another study, Tsikritzis et al. doped PC_70_BM with ultra−thin Bi_2_Te_3_ flakes [205]. The optimal doping of PC_70_BM with Bi_2_Te_3_ flakes was 2% *v*/*v*, which resulted in a PCE of 18.0%. After that, they also used Bi_2_Te_3_ flakes as the interlayer. They formed two spin coatings of the Bi_2_Te_3_ flakes dispersion onto the PC_70_BM, leading to a PCE of 18.6%. Finally, they combined two engineering approaches, obtaining a PCE of 19.46%. This was the highest record for inverted PSCs at that time. In another study, Rueda−Delgado et al. fabricated C_60_ interlayers and deposited either at the interface between SnO_2_ and MAPbI_3−x_Cl_x_ absorber layer in a n−i−p architecture. C_60_ interlayers reduced the charge carrier accumulation at the interface. The final optimal PCE was 17.3% [206]. Compared with the traditional upright PSCs, the inverted PSCs still has a certain gap.

### 5.2. Organic Small−Molecule Materials

Novel organic small−molecule materials have been developed as ETL even though C_60_ and its derivatives have been comprehensively applied as electron transport materials for typical configuration of inverted PSCs, as is shown in Table 18 [207,208,209,210]. In 2017, Wu et al. synthesized three perylene diimides (X−PDI, X = H, F, or Br) as the substitution of PCBM in inverted PSCs [207]. With the ZnO nanoparticles cathode buffer layer, the Br−PDI based devices demonstrated an outstanding efficiency of 10.5% because of the high direct current conductivity and electron mobility of Br−PDI. Gu et al. adopted a specific micro−molecule organic (TDTP) to replace PCBM as ETL for inverted devices and achieved an outstanding efficiency of 18.2% due to strengthen interaction of TDTP and perovskite film [208]. In 2018, Jiang et al. synthesized a 3D type perylenediimide−based molecule (TPE−PDI4) with excellent electron mobility and solution workability and applied it as ETL for inverted devices. Electron transport performance of TPE−PDI4 is higher than that of PCBM under the same conditions. Furthermore, the TPE−PDI4 has excellent water resistance, which could protect the perovskite layer more effectively and obtain a PCE up to 18.78% [209]. Wu et al. employed two ITCPTC−based n−type π−conjugated small molecules, namely, ITCPTC−Se and ITCPTC−Th, as electron transport materials for inverted devices [210]. Devices with ITCPTC−Th materials illustrated an excellent PCE of 17.11% due to smoother morphology and the better electron transporting properties of ITCPTC−Th. In addition, the devices with ITCPTC−Th interlayer at perovskite/C_60_ ETL interface achieved the high PCE nearly 19%. Organic small−molecule material can also tune the energy gap and increase the electron affinity. The small−molecule material is promising in the PSCs [211]. However, there are relatively few researches on organic small−molecule materials. Moreover, the efficiency of the devices cannot be compared with that of fullerenes materials due to the constraints of the solvent and energy level matching of the materials. Therefore, the development of efficient, inexpensive organic materials for electron transport remains a huge challenge in inverted structure.

### 5.3. Inorganic Electron Transport Materials

The removal of organic components and the application of inorganic ETL are effective methods to improve thermally stable of inverted PSCs. The n−type semiconductor materials were surveyed and employed to replace organic ETL for inverted devices, such as ZnO and CdS, as is shown in Table 19 [212,213,214,215,216,217,218,219,220,221,222,223,224,225,226]. In 2016, You et al. applied the n−type ZnO nanoparticles as ETL to resist the water and oxygen degradation [212]. With the application of ZnO nanoparticles, the devices retained about 90% of initial efficiency in the atmosphere at room temperature. Li et al. prepared a ribboned compound, Bi_2_S_3_, as the ETL via a thermal evaporation method for the inverted devices [213]. Moreover, the ambient storage stability was enhanced due to the compact and smooth amorphous because of the hydrophobicity and sealed packaging of absorption film with compact Bi_2_SIn 2017, Tan et al. introduced the CdSe quantum dots (QDs) as ETL and LiF as buffer to achieve prominent photoelectric properties of inverted PSCs [214]. The introduction of CdSe QDs/LiF double layer could effectively promote the separation and transfer of charge, obtaining a high PCE of 15.1%. In 2018, Hu et al. fabricated a CeO_x_ layer as ETL for the inverted PSCs via a solution process [215]. The dense CeO_x_ could prevent moisture from corroding the perovskite film and the metal electrode, leading to an enhanced stability with more than 90% of initial efficiency under 200 h of illumination. In 2019, Hossain et al. developed a scalable low−temperature TiO_2_ ETL based on presynthesized crystalline nanoparticle (np−TiO_2_). Nb^5+^ doping increased the conductance through the np−TiO_2_ amd reduced the series resistance. The optimal PCE they obtained was 19.5% [227]. In 2022, Eliwi et al. designed bilayer SnO_2_ QDs ETLs. They found that the bilayer ETL composed of lithium−doped compact SnO_2_ (c(Li)−SnO_2_) at the bottom and potassium−capped SnO_2_ nanoparticle layers (NP−SnO_2_) at the top enhanced charge transport properties of PSCs, and significantly reduced the degree of ion migration. This resulted optimal PCE reached 20.4%, and strongly reduced J−V hysteresis for PSCs [228].

In 2021, Jiang et al. adopted a Nb_2_O_5_ layer as ETL and the polyacetylene derivatives doped perovskite as a absorber layer to fabricate the inverted PSCs, acquiring an outstanding efficiency of 20.41% and the prominent stability of devices [218]. Yang et al. synthesized the In_2_O_3_ and Sn:In_2_O_3_ nanoparticles as ETL in inverted PSCs through a low temperature preparation method, which led to an impressive PCE up to 20.65% due to the enhanced built−in field, efficient electron extraction and decreased charge recombination [219]. All−inorganic PSCs can be prepared by combining inorganic ETL with inorganic HTL and inorganic perovskite active layer. Zhang et al. prepared the carbide−metal oxides (C−MOXs) ETL on the inorganic perovskite of CsPbI_2_Br film to improve long periods of stability, maintaining over 90% of initial PCE at 85 °C without illumination and continuous under 45 °C with 1000 h of illumination [216].

## 6. All−Inorganic Perovskite Solar Cells

Though the PCE of inverted PSCs has exceeded 23%, it is expected to replace traditional crystalline silicon solar cells. However, the stability of hybrid PSCs has seriously hindered their commercialization. The key element influencing stability is existence of organic cations that decompose easily under high temperature and ultraviolet light. Therefore, the preparation of devices by inorganic materials is the key to solving the problem. Inorganic perovskite materials formed by substituting organic groups with inorganic cations (Cs^+^) are ideal light absorption materials for the construction of stable PSCs because of their excellent photothermal stability and comparable photoelectric properties to hybrid perovskite materials, including CsPbI_3_, CsPbI_2_Br, CsPbIBr_2_ and CsPbBr. Combined with inorganic charge transport materials, the design and preparation of all−inorganic PSCs are expected to break through the bottleneck of PSCs, as is shown in Table 20 [229,230,231,232,233,234,235,236,237], and the development of high performance all−inorganic inverted PSCs is desirable.

Among the above inorganic perovskite materials, CsPbI_3_ has a narrow band gap and matched energy level, which is one of the ideal choices for building high efficiency devices [229]. However, CsPbI_3_ has the problem of unstable perovskite phase at room temperature, and it is easy to decay in atmosphere environment. CsPbIBr_2_ and CsPbBr_3_ are more stable in an atmosphere environment. However, the CsPbIBr_2_ and CsPbBr_3_ with wide band gap are not conducive to the absorption and utilization of sunlight and the improvement of device performance. For instance, Yang et al. adopted Cs−doped NiO_x_ as hole transport materials and inserted a layer of N749 at the Cs−NiO_x_/CsPbIBr_2_ interface for better energy alignment, as is shown in Figure 8a,b [231]. As a result, the device with the construction of Cs−NiO_x_/N749/CsPbIBr_2_/PCBM/BCP/Ag illustrated an efficiency of 9.49% and, prominently, moisture stability with over 86% of the initial PCE for 1000 h with 65% relative humidity. Therefore, CsPbI_2_Br is the most commonly used perovskite material. Furthermore, the commonly used electron−transport materials are fullerenes and their derivatives in all−inorganic inverted devices. In 2018, Liu et al. used NiO_x_ thin film and ZnO@C_60_ bilayer as HTL and ETL, respectively [232]. PSCs with structure of NiO_x_/CsPbI_2_Br/ZnO@C_60_/Ag exhibited efficient extraction of photoexcited charge, low energy loss and outstanding stability. Zhang et al. fabricated a C−TiO_2_ ETL and a Sb electrode on the surface of CsPbI_2_Br absorption layer in order to remove organic components in the devices [223]. Consequently, the designed devices demonstrated a high efficiency of 14.8% and the excellent stability with more than 90% of original efficiency at 60 °C for 1000 h. Chen et al. tailored the crystallinity of CsPbI_2_Br through F^−^ doping, and prepared the ETL with gradient energy alignment, including NiO_x_/Zn:CuGaO_2_ HTL and TiO_2_/PC_61_BM/ZnO ETL, to construct the high performance device with the structure of NiO_x_/Zn:CuGaO_2_/F−CsPbI_2_Br/TiO_2_/PC_61_BM/ZnO [236]. Consequently, the fabricated device illustrated an excellent efficiency of 15.1% with excellent thermal and operational stability. Wang et al. passivated the CsPbI_x_Br_3−x_ with a Lewis base small molecule (6TIC−4F) to decrease non−radiative recombination, which gave rise to a highest efficiency of 16.1% with enhanced photostability, as is shown in Figure 8c,d [237].

## 7. Conclusions and Perspectives

The inverted PSCs emerged in with the improvement of traditional upright devices. Properties of inverted devices have also made significant progress in recent years, with more than 24% efficiency being achieved. The architecture and the preparation process of inverted PSCs have been systematically investigated, including electron transport layer, perovskite absorption layer, interface modification layer, doping, etc. With the research of underlying mechanism, the stability of devices has been significantly enhanced in recent years. Furthermore, the photoelectric performance of the device is getting better and better with development of all−inorganic inverted PSCs.

Up to now, the highest PCE of inverted devices has achieved 24.3%, which will catch up with the highest PCE of 25.5% for the traditional upright PSCs, and the hysteresis effect can be reduced with the application of fullerene derivatives ETL, leading to valid and reliable tested results. In addition, most of preparing technology for the inverted PSCs involves a low−temperature preparation process, which is beneficial to reduction of production costs and the preparation of flexible PSCs. Reports of superior PCE, long periods of stability and inexpensive for preparation make the inverted PSCs more competitive during the process of solar cells commercialization. Moreover, the inverted PSCs are more suitable for 2T perovskite−silicon tandem solar cells than the traditional upright PSCs [238]. Because the simulation studies determined strong parasitic absorption loss in UV/Visible region of the spectra by conventional Spiro−OMeTAD, they also show inclusion of these as hole transport material is undesirable in an n−i−p configurations. However, the stability, large area and environmental protection of inverted PSCs are still the bottlenecks restricting its industrialization at present. On the basis of understanding the decomposition mechanism of perovskite and charge transport materials, as well as ion migration and interfacial non−radiative recombination between materials, the improvement of device stability can be carried out in the development and design of better device packaging technology, novel charge transport materials, optimization of interface modification and regulation of interface charge transfer. However, the fabrication of high quality large−area devices remains a serious challenge due to poor property of film forming for perovskite and charge transport material films with large area. As the device area increases, (1) The uniformity of perovskite film will decrease, and the defects will increase. (2) The effective light area associated with device structure will be reduced, resulting in lower J_sc_. (3) It is related to the design of the series−parallel structure and the component technology, resulting in an increase in the series resistance of the components and a decrease in the parallel resistance. Now, there are many processing technologies, such as solvent treatment, moisture assisted growth, hot spin−coating and additives, which have been confirmed as an effective way. Exploring new methods or combining these existing technologies to further improve the film quality should be our next direction. Moreover, encapsulation technology is another significant component necessary to enhance the long stability of inverted PSCs. Fullerenes can forbid ion movement and enhance the charge transfer to reduce the hysteresis. A fullerene/metal oxide such as PCBM/ZnO bilayer system shows promise in terms of reduced hysteresis and improved highly stability. Moreover, the small−molecule materials we introduced in front of the review are continuously developed by researchers. They play a significant role in enhancing the performance of the inverted PSCs. So, we think we can pay attention to the small−molecule materials.

In addition, more stable and environmentally friendly PSCs without lead or with less lead can be developed by replacing or reducing the content of Pb in perovskite film with non−toxic metal elements, such as Sn and Bi. Therefore, if the development of new methods, new materials or new ideas can solve the issues of efficiency, stability, large area, inexpensive and environmental protection during the process of commercialization, it will effectively promote the industrial application of inverted PSCs.

## Figures and Tables

**Figure 1 ijms-23-11792-f001:**
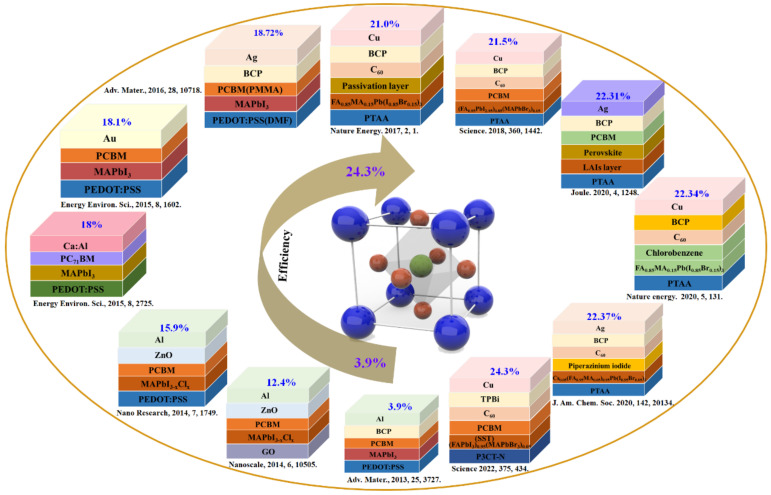
Photoelectric performance and the structure of typical inverted PSCs [30,31,32,33,34,35,36,37,38].

**Figure 2 ijms-23-11792-f002:**
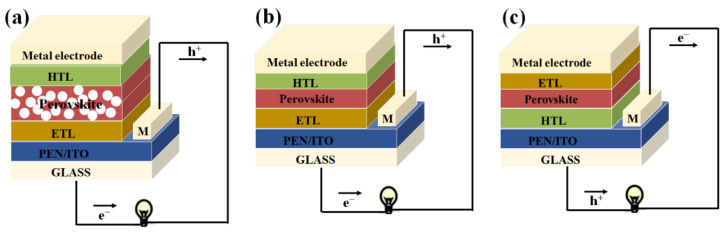
The representative structures of PSCs devices: (**a**) mesoporous upright structure; (**b**) upright structure; (**c**) inverted structure.

**Figure 3 ijms-23-11792-f003:**
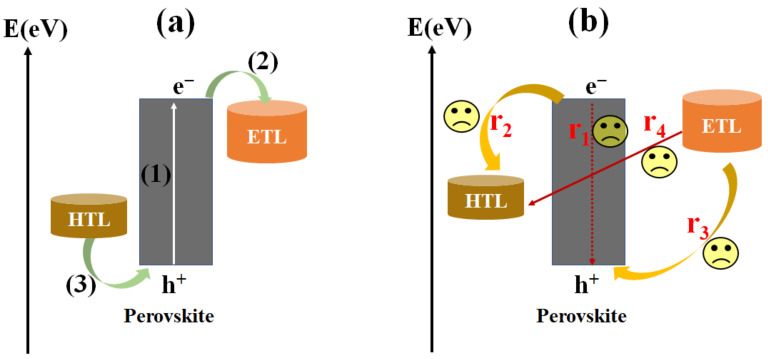
The schematic of energy structure and carriers transfer processes in inverted devices: (**a**) Desirable charge transfer processes [41]. (**b**) Undesirable charge recombination processes [41].

**Figure 4 ijms-23-11792-f004:**
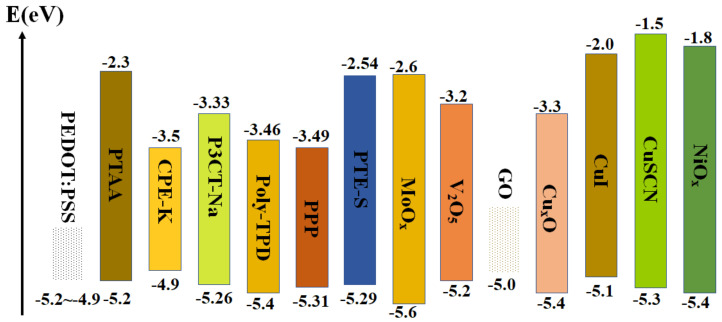
Energy levels of different HTM in inverted PSCs, including conductive polymers materials, organic small−molecule materials and inorganic p−type semiconductor materials [44,45].

**Figure 5 ijms-23-11792-f005:**
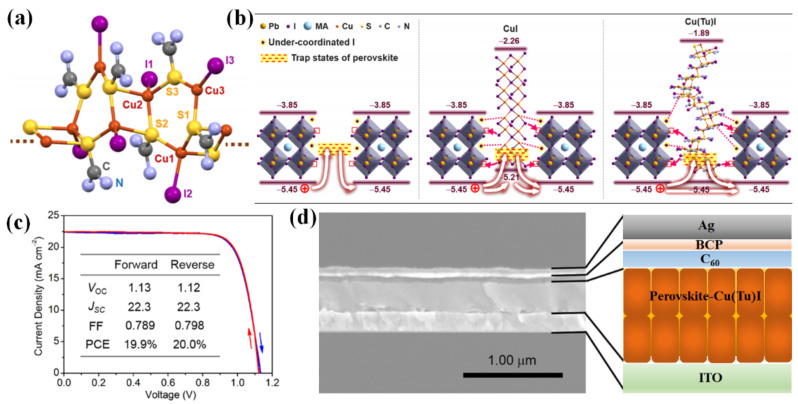
(**a**) Crystal texture of Cu(Tu)I [163]; (**b**) Sketch map of probable cause for the trap state suppression [163]; (**c**) Greatest J−V graph of inverted devices on the basis of Cu(Tu)I [163]; (**d**) Sectional view and device structure of inverted PSCs [163].

**Figure 6 ijms-23-11792-f006:**
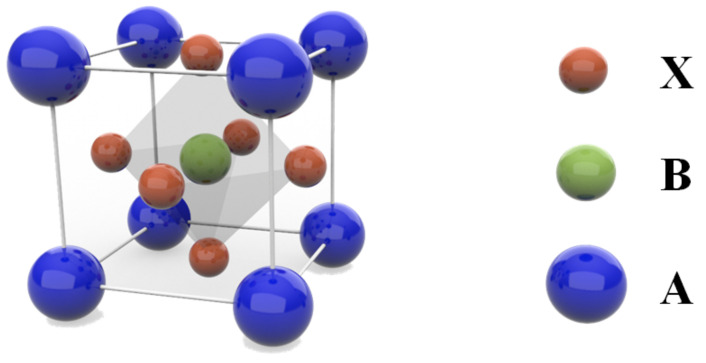
Crystal structure of perovskite absorption material ABX_3_ (A: CH_3_NH_3_^+^, CH_2_(NH_2_)_2_^+^, or Cs^+^; B: Pb^2+^ or Sn^2+^; X: Cl^−^, Br^−^ or I^−^).

**Figure 7 ijms-23-11792-f007:**
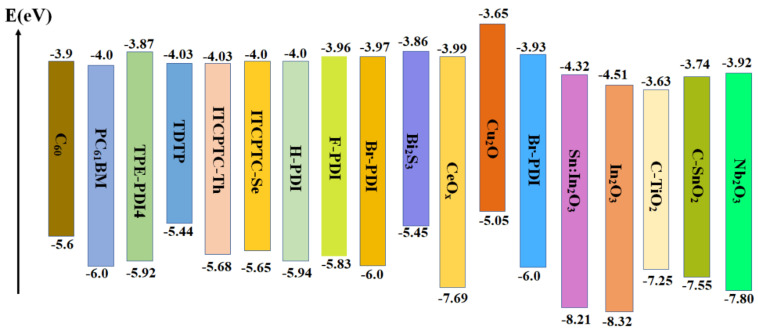
Energy levels of electron transport materials in inverted PSCs, including C_60_ and its derivatives as well as other electron transport materials.

**Figure 8 ijms-23-11792-f008:**
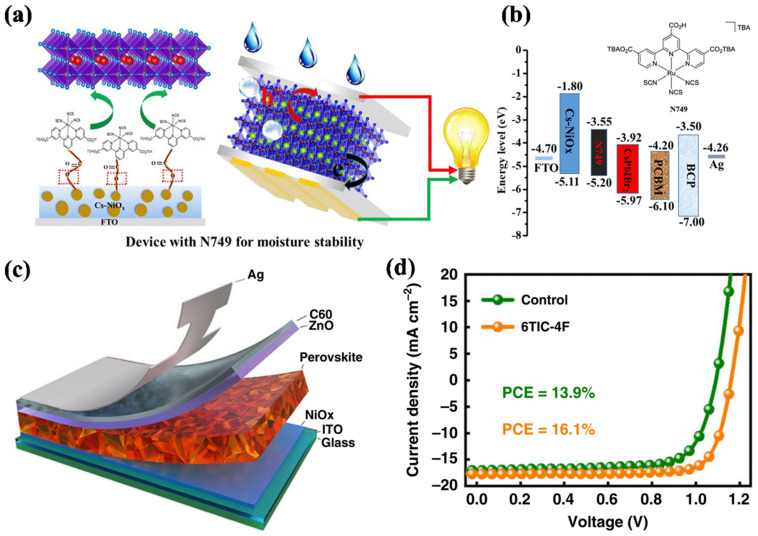
(**a**) Device with N749 for moisture stability [231]; (**b**) Energy structure of PSCs with the construction of Cs−NiO_x_/N749/CsPbIBr_2_/PCBM/BCP/Ag [231]; (**c**) Device structure of devices with the construction of NiO_x_/CsPbI_x_Br_3−x_/ZnO/C_60_/Ag [237]; (**d**) J−V graph of PSCs [237].

**Table 1 ijms-23-11792-t001:** Performances of several representative inverted PSCs on the basis of pure PEDOT:PSS.

Structure of PSCs	Area	V_oc_ (V)	J_sc_ (mA/cm^2^)	FF	PCE (%)	Ref.
PEDOT:PSS/MAPbI_3_/PCBM/BCP/Al	0.06 cm^2^	0.60	10.32	0.63	3.9	[30]
PEDOT:PSS/MAPbI_3_/PCBM/LiF/Al	25 cm^2^	0.93	18.0	0.77	12.8	[46]
PEDOT:PSS/MAPbI_3_/PCBM/Bis−C_60_/Ag	3.14 mm^2^	1.06	19.1	0.77	15.7	[47]
PEDOT:PSS/MAPbI_3_/PCBM/TiO_2_/Al	11.8 mm^2^	0.994	19.8	0.72	14.25	[48]
PEDOT:PSS/MAPbI_3−x_Cl_x_/PCBM/ RhB101/LiF/Ag	0.11 cm^2^	1.11	20.11	0.81	18.0	[49]
PEDOT:PSS/MAPbI_3_/PCBM/Au	0.16 cm^2^	1.1	20.9	0.79	18.1	[37]

**Table 2 ijms-23-11792-t002:** Performances of several representative inverted PSCs on the basis of doping PEDOT:PSS with different substances.

Structure of PSCs	Area	V_oc_ (V)	J_sc_ (mA/cm^2^)	FF	PCE (%)	Ref.
PEDOT:PSS−PEI/MAPbI_3_/PCBM/Al	0.06 cm^2^	0.982	16.7	0.705	11.7	[50]
PEDOT:PSS−GO/MAPbI_3_/PCBM/Al	~	0.96	17.96	0.76	13.1	[51]
PEDOT:PSS−GO/MAPbI_3_/PCBM/BCP/Ag	0.09 cm^2^	0.90	20.01	0.79	14.2	[52]
PEDOT:PSS−PFI/FA_0.6_MA_0.4_Sn_0.6_Pb_0.4_I_3_/PCBM/BCP/Ag	~	0.783	27.22	0.744	15.85	[53]
PEDOT:PSS−RbCl/MA_0.7_FA_0.3_Sn_0.6_Pb(I_0.9_Br_0.1_)_3_/PCBM/C_60_/BCP/Ag	~	1.0	22.41	0.824	18.3	[54]
PEDOT:PSS−CsI/MAPbI_3_/PCBM/Ag	~	1.084	22.58	0.83	20.22	[55]
PEDOT:PSS−Sodium Citrate/MAPbI_3_(Cl)/PCBM/BCP/Ag	0.09 cm^2^	1.134	21.62	0.75	18.39	[56]
PEDOT:PSS−CuSCN/MAPbI_3_/ PCBM/C_60_/LiF/Al	~	1.02	19.10	0.785	15.3	[57]
PEDOT:PSS−GeO_2_/MAPbI_3−x_Cl_x_/ PCBM/Ag	7.25 mm^2^	0.96	21.55	0.74	15.15	[58]
PEDOT:PSS−Ag/MAPbI_3−x_Cl_x_/PCBM/ Bphen/Ag	7.25 mm^2^	0.93	21.51	0.79	15.75	[59]
PEDOT:PSS−PEO/MAPbI_3_/PCBM/Al	0.16 cm^2^	0.88	23.42	0.801	16.52	[60]
PEDOT:PSS−F4−TCNQ/MAPbI_3−x_Cl_x_ /PCBM/BCP/Ag	7.25 mm^2^	1.02	21.93	0.77	17.22	[61]
PEDOT:PSS−NaCl/MAPbI_3−x_Cl_x_/PCBM/RhB101/LiF/Ag	0.11 cm^2^	1.08	20.5	0.819	18.1	[62]
GHJ/MAPbI_3_/PCBM/C_60_/LiF/Al	~	1.02	22.98	0.77	18.0	[64]

**Table 3 ijms-23-11792-t003:** Performances of several representative inverted PSCs on the basis of the modified PEDOT:PSS.

Structure of PSCs	Area	V_oc_ (V)	J_sc_ (mA/cm^2^)	FF	PCE (%)	Ref.
DMF−PEDOT:PSS/MAPbI_3−x_Cl_x_/PCBM/RhB101/LiF/Ag	0.11 cm^2^	1.08	17.44	0.68	12.9	[65]
DMF−PEDOT:PSS /MAPbI_3_/PCBM(PMMA)/BCP/Ag	0.09 cm^2^	1.02	22.38	0.82	18.72	[32]
DMF−PEDOT:PSS/MAPbI_3_/PCBM/BCP/Ag	9 mm^2^	1.048	21.1	0.76	16.8	[66]
DMSO−PEDOT:PSS/MAPbI_3_/PCBM/ BCP/Ag	1.8 mm^2^	0.92	22.76	0.80	16.7	[67]
EMIC−PEDOT:PSS/MAPbI_3_/Passivation layer/C_60_/BCP/Ag	1 cm^2^	1.08	23.81	0.78	20.06	[68]
DA−PEDOT:PSS/MAPbI_3−x_Cl_x_/PCBM/BCP/Ag	~	1.08	19.4	0.78	16.4	[69]
DA−PEDOT:PSS/MA_x_FA_3−x_PbI_3−x_Br_x_/PCBM/PN_4_N/Ag	~	1.08	22.0	0.775	18.5	[70]
Oxidized PEDOT:PSS/MAPbI_3−x_Cl_x_/PCBM/ RhB 101/LiF/Ag	11 mm^2^	1.07	21.6	0.82	18.8	[71]
PEDOT−MeOH:PSS/MAPbI_3_/PCBM/ BCP/Ag	3.14 mm^2^	1.01	15.83	0.60	9.56	[72]
Urea−PEDOT:PSS/MAPbI_3_/PCBM/ RhB 101/Ag	0.16 cm^2^	1.03	22.57	0.809	18.8	[73]
SBS−PEDOT:PSS/MA_0.8_FA_0.2_PbI_3−x_Cl_x_/PCBM/Ag	~	1.08	21.57	0.833	19.41	[74]

**Table 4 ijms-23-11792-t004:** Performances of several representative inverted PSCs on the basis of interfacial modified PEDOT:PSS.

Structure of PSCs	Area	V_oc_ (V)	J_sc_ (mA/cm^2^)	FF	PCE (%)	Ref.
PEDOT:PSS/Cross−linked interlayer/MAPbI_3−x_Cl_x_/PCBM/Al	7.5 mm^2^	0.99	18.07	0.73	13.06	[75]
PEG−PEDOT:PSS/FASnI_3_/PCBM/BCP/Ag	~	0.37	22.06	0.627	5.12	[76]
PEDOT:PSS/PEG/MAPbI_3_/PCBM/Ag	0.06 cm^2^	0.79	23.02	0.61	12.56	[77]
PEDOT:PSS/NPB/MAPbI_3_/PCBM/ BCP/Ag	0.1225 cm^2^	1.05	22.46	0.78	18.4	[78]
PEDOT:PSS/C3−SAM/MAPbI_3−x_Cl_x_/ PCBM/ZnO NPs/Ag	0.055 cm^2^	0.89	18.9	0.69	11.6	[79]
PEDOT:PSS/GO/MAPbI_3_/PCBM/Ag	1 cm^2^	0.985	21.9	0.71	15.34	[80]

**Table 5 ijms-23-11792-t005:** Performances of several representative inverted PSCs on the basis of the composite HTL.

Structure of PSCs	Area	V_oc_ (V)	J_sc_ (mA/cm^2^)	FF	PCE (%)	Ref.
PEDOT:PSS/V_2_O_5_/MAPbI_3_/C_60_/ BCP/Ag	0.10 cm^2^	1.05	0.212	0.785	17.5	[81]
NiPcS_4_−PEDOT:PSS/MAPbI_3_/C_60_/PCBM/BCP/Ag	0.08 cm^2^	1.08	23.01	0.77	18.9	[82]
CNTs−PEDOT:PSS/MAPbI_3_/PCBM/Ag	~	1.04	20.35	0.754	16.0	[83]
OCNRs−PEDOT:PSS/MAPbI_3_/ PCBM/BCP/Ag	~	1.01	22.76	0.804	19.02	[84]
MoO_x_−PEDOT:PSS/MAPbI_3_/PCBM/ ZnO/Al	0.06 cm^2^	1.08	22.78	0.804	19.64	[85]
CuI/PEDOT:PSS/(FASnI_3_)_0.6_(MAPbI_3_)_0.4_/C_60_/BCP/Cu	0.1 cm^2^	0.75	28.5	0.737	15.75	[86]
PEDOT:PSS/Graphene quantum dots/MAPbI_3_/PCBM/Ag	0.1 cm^2^	1.00	21.41	0.753	16.16	[87]
PEDOT:PSS/PTAA/MAPbI_3−x_Cl_x_/ PCBM/Ag	0.11 cm^2^	1.07	21.38	0.826	19.04	[88]

**Table 6 ijms-23-11792-t006:** Performances of several representative inverted PSCs on the basis of the PTAA materials with different structures.

	Structure of PSCs	Area	V_oc_ (V)	J_sc_ (mA/cm^2^)	FF	PCE (%)	Ref.
Pure PTAA	PTAA/MAPbI_3_/PCBM/C_60_/ BCP/Al	7.25 mm^2^	1.07	22.0	0.768	18.1	[89]
	PTAA/MAPbI_3_/PCBM/PFN/Al	4 mm^2^	1.01	20.24	0.767	15.67	[90]
	PTAA/(FA_0.95_PbI_2.95_)_0.85_(MAPbBr_3_)_0.5_/PCBM/C_60_/BCP/Cu	~	1.21	22.5	0.790	21.5	[34]
Doping PTAA	PTAA−F4−TCNQ/MAPbI_3_/PCBM/C_60_/BCP/Al	~	1.09	21.6	0.740	17.5	[91]
	PTAA−NPB/MAPbI_3_/PCBM/Ag	6.25 mm^2^	1.14	22.6	0.784	20.15	[92]
	PTAA−CuSCN/MAPbI_3_/PCBM/ BCP/Ag	0.1 cm^2^	1.12	21.92	0.750	18.16	[93]
Modified PTAA	rGO/PTAA/MAPbI_3_/PCBM/ BCP/Ag	1.02 cm^2^	1.09	20.3	0.777	17.20	[94]
	Spiro−OMeTAD−PTAA/CsPbI_2_Br/ZnO:C_60_/Ag	4 mm^2^	1.14	14.3	0.764	12.52	[95]
	PTAA/F8BT/MAPbI_3_/PCBM/Bphen/Ag	0.045 cm^2^	0.98	21.27	0.717	14.88	[96]
	PTAA/F4−TCNQ/MAPbI_3_/ PCBM/BCP/Ag	0.09 cm^2^	1.10	22.6	0.811	19.7	[97]
	PTAA/CPEs/(FAPbI_3_)_0.83_(MAPbBr_3_)_0.17_/LiF/C_60_/Ag	1 cm^2^	1.11	22.18	0.747	18.38	[98]
Interface engineering	PTAA/MAPbI_3_/C_60_/BCP/Cu	6 mm^2^	1.05	22.7	0.80	19.0	[99]
	PTAA/(FAPbI_3_)_0.9_(MAPbBr_3_)_0.1_/ PCBM/BCP/Ag	0.05 cm^2^	1.09	23.87	0.801	19.51	[100]
	PTAA/Cs_0.05_(MA_0.17_FA_0.83_)_0.95_ Pb(I_0.83_Br_0.17_)_3_/PCBM/BCP/Cu	0.09 cm^2^	1.08	22.74	0.78	19.17	[101]
	PTAA/MAPbI_3_/PCBM/BCP/Ag	6 mm^2^	1.14	23.26	0.815	21.6	[102]

**Table 7 ijms-23-11792-t007:** Performances of several representative inverted PSCs on the basis of the different CPE HTL structures.

Structure of PSCs	Area	V_oc_ (V)	J_sc_ (mA/cm^2^)	FF	PCE (%)	Ref.
CPE−K/MAPbI_3−x_Cl_x_/PCBM/Al	3.30 mm^2^	0.89	20.1	0.77	12.51	[105]
TB(MA)/MAPbI_3_/PCBM/C_60_/BCP/Ag	0.10 cm^2^	1.08	23.45	0.78	19.76	[106]
BF−NH_3_/MAPbI_3_/PC_61_BM/PEI/Ag	~	1.05	20.1	0.84	17.71	[107]
TB(K)/FA_0.85_MA_0.15_Pb(Br_0.15_I_0.85_)_3_/ PCBM/C_60_/BCP/Ag	0.06 cm^2^	1.1	22.72	0.80	20.1	[108]

**Table 8 ijms-23-11792-t008:** Performances of several representative inverted PSCs on the basis of the different polyelectrolyte HTL structures.

Structure of PSCs	Area	V_oc_ (V)	J_sc_ (mA/cm^2^)	FF	PCE (%)	Ref.
P3CT−Na/MAPbCl_3−x_I_x_/PC_61_BM/C_60_	~	1.07	21.14	73.0	16.6	[111]
P3CT−Na(GD)/MAPbI_3_/PCBM/ZnO/Al	~	1.06	22.8	80.8	19.5	[112]
P3CT−Rb/MAPbCl_3−x_I_x_/C_60_/BCP/Ag	~	1.14	21.67	82.8	20.52	[113]
P3CT−ED/MAPbI_3_/PC_61_BM/ZnO/Al	0.06 cm^2^	1.08	23.3	80.9	20.5	[114]

**Table 9 ijms-23-11792-t009:** Performances of several representative inverted PSCs on the basis of the different Poly−TPD HTL structures.

Structure of PSCs	Area	V_oc_ (V)	J_sc_ (mA/cm^2^)	FF	PCE (%)	Ref.
Poly−TPD/MAPbI_3_/PCBM/C_60_/BCP/Ag	~	1.1	22.0	0.697	15.3	[115]
Poly−TPD/MAPbI_3_/C_60_/BCP/Ag	0.1 cm^2^	1.04	23.2	0.754	18.19	[116]
Poly−TPD/MAPbI_3_/PCBM/BCP/Ag	9 mm^2^	1.07	20.08	0.75	16.11	[117]
Poly−TPD/MAPbI_3_/PCBM/LiF/Al	0.045 cm^2^	1.07	21.8	0.737	17.2	[118]
Poly−TPD/PFN−I/Cs_0.05_FA_0.79_MA_0.16_PbI_2.4_Br_0.6_/PFN−I/PC_61_BM/BCP/Ag	~	1.13	22.47	0.81	20.47	[119]

**Table 10 ijms-23-11792-t010:** Performances of several representative inverted PSCs on the basis of other conductive polymers HTL.

Structure of PSCs	Area	V_oc_ (V)	J_sc_ (mA/cm^2^)	FF	PCE (%)	Ref.
ITO/polythiophene/MAPbI_3_/C_60_/BCP/Ag	~	0.99	20.3	0.774	15.4	[120]
ITO/PPP/MAPbI_3_/C_60_/BCP/Ag	~	1.03	21.6	0.75	16.5	[121]
PVK/MAPbI_3_/PCBM/Ag	4 mm^2^	0.96	21.9	0.75	15.8	[122]
XSln847/MAPbI_3_/PCBM/Ag	~	1.08	22.34	0.71	17.16	[123]
poly−1/MAPbI_3_/PC_61_BM/BCP/Ag	0.1 cm^2^	1.01	23.2	0.71	16.5	[124]

**Table 11 ijms-23-11792-t011:** Performances of several representative inverted PSCs on the basis of organic small−molecule HTL.

Structure of PSCs	Area	V_oc_ (V)	J_sc_ (mA/cm^2^)	FF	PCE (%)	Ref.
TPASB/MAPbI_3_/PCBM/Al	4 mm^2^	1.05	20.8	0.80	17.6	[126]
TAPC/MAPbI_3_/PCBM/Ag	4 mm^2^	1.04	22.32	0.81	18.8	[127]
VB−MeO−FDPA/MAPbI_3−x_Cl_x_/ PCBM/Ag	0.045 cm^2^	1.15	20.89	0.78	18.7	[128]
NPB/MAPbI_3_/PCBM/PDI−Br/Ag	~	1.11	22.92	0.78	19.96	[129]
DFH/MA_0.9_FA_0.1_PbI_3−x_Cl_x_/C_60_/ BCP/Ag	~	1.10	22.6	0.83	20.6	[130]
MPA−BTTI/CsFAMA/C_60_/BCP/Ag	~	1.12	23.23	0.814	21.17	[131]
TPE−S/CsPbI_2_Br/PCBM/ZnO/Ag	~	1.26	15.6	0.785	15.4	[132]
C_8_−DPNDF/MAPbI_3_/C_60_/BCP/Ag	9 mm^2^	1.06	21.05	0.784	17.5	[133]
TFM/CsFAMA/C_60_/BCP/Ag	7.5 mm^2^	22.7	0.97	0.73	16.03	[134]
H−Pyr/MAPbI_3_/PCBM/Ag	0.1225 cm^2^	1.04	22.26	0.741	17.09	[135]
XY1/CsFAMA/C_60_/BCP/Cu	1 cm^2^	1.11	22.2	0.762	18.78	[136]
m−MTDATA/Cs_0.05_(FA_0.85_MA_0.15_)_0.95_ Pb(I_0.85_Br_0.15_)_3_/C_60_/BCP/Cu	0.16 cm^2^	1.04	22.5	0.78	18.12	[137]
TPAC3M/MAPbI_3_/PC_61_BM/ZnO/Al	4 mm^2^	1.00	22.11	0.78	17.54	[138]

**Table 12 ijms-23-11792-t012:** Performances of several representative inverted PSCs on the basis of NiO_x_ materials.

	Structure of PSCs	Area	V_oc_ (V)	J_sc_ (mA/cm^2^)	FF	PCE (%)	Ref.
Pure NiO_x_	NiO_x_/MAPbI_3_/PCBM/Au	~	0.882	16.27	0.635	9.11	[140]
	NiO_x_/MAPbI_3_/PCBM/LiF/Al	~	1.06	20.2	0.813	17.3	[141]
	NiO_x_/MAPbI_3_/PCBM/Ag	~	1.09	17.93	0.738	14.42	[142]
	NiO_x_/MAPbI_3−x_Cl_x_/PC_61_BM/ AZO/Ag	0.19 cm^2^	1.08	20.33	0.69	15.2	[143]
	NiO_x_/MAI/PCBM/BCP/Ag	~	0.99	22.92	0.803	18.15	[144]
	np−NiO_x_/(FAPbI_3_)_0.85_−(MAPbBr_3_)_0.15_/PCBM/ZnO/Ag	0.09 cm^2^	1.076	22.76	0.78	19.1	[145]

**Table 13 ijms-23-11792-t013:** Performances of several representative inverted PSCs on the basis of doping NiO_x_ HTL.

	Structure of PSCs	Area	V_oc_ (V)	J_sc_ (mA/cm^2^)	FF	PCE (%)	Ref.
Doping NiO_x_	Cu:NiO_x_/MAPb(I_0.8_Br_0.2_)_3_/ PC_61_BM/C_60_/Ag	0.314 cm^2^	1.12	18.83	0.73	15.4	[146]
	Li_0.05_Mg_0.15_Ni_0.8_O/MAPbI_3_/ PCBM/Ti(Nb)O_x_/Ag	1.02 cm^2^	1.08	20.4	0.827	18.3	[147]
	Cs:NiO_x_/MAPbI_3_/PC_61_BM/ ZrAcac/Ag	10 mm^2^	1.12	21.77	0.793	19.35	[148]
	Ag:NiO_x_/MAPbI_3_/PCBM/ BCP/Ag	4 mm^2^	1.07	19.42	0.79	16.4	[149]
	Y:NiO_x_/MAPbI_3_/PC_61_BM/Au	0.08 cm^2^	1.0	23.82	0.68	16.31	[150]
	F6TCNNQ:NiO_x_/CsFAMA/ PCBM/ZrAcac/Ag	~	1.12	23.18	0.803	20.86	[151]
	K:NiO_x_/Cs_0.05_FA_0.81_MA_0.14_Pb (Br_0.15_I_0.85_)_3_/PC_61_BM/TIPD/Ag	4 mm^2^	1.13	20.53	0.74	17.05	[152]

**Table 14 ijms-23-11792-t014:** Performances of several representative inverted PSCs on the basis of interfacial modified NiO_x_.

	Structure of PSCs	Area	V_oc_ (V)	J_sc_ (mA/cm^2^)	FF	PCE (%)	Ref.
Interface modification	NiO/meso−Al_2_O_3_/MAPbI_3_/PCBM/BCP/Ag	0.09 cm^2^	1.04	18.0	0.72	13.5	[153]
	NiO_x_/PTAA/FA_1−x_MA_x_Pb (I_3−y_Br_y_)/PCBM/BCP/Au	~	1.02	20.8	0.783	16.7	[154]
	NiO_x_@KCl/CsFAMA/ PCBM/ZrAcac/Ag	0.1 cm^2^	1.15	22.21	0.795	20.96	[155]
	NiO_x_/PTAA/(MAPbI_3_)_0.95_(MAPbBr_2_Cl)_0.05_/PCBM/BCP/Ag	~	1.19	22.23	0.817	21.56	[156]
	NiO_x_/F2HCNQ/CsFAMA/PCBM/BCP/Ag	36.1 cm^2^	1.14	23.44	0.828	22.13	[157]

**Table 15 ijms-23-11792-t015:** Performances of several representative inverted PSCs on the basis of the Cu−based HTL structures.

	Structure of PSCs	Area	V_oc_ (V)	J_sc_ (mA/cm^2^)	FF	PCE (%)	Ref.
CuSCN	CuSCN/MAPbI_3−x_Cl_x_/PCBM/Ag	0.07 cm^2^	0.68	8.8	0.635	3.8	[158]
	CuSCN/MAPbI_3_/C_60_/BCP/ Ag	0.10 cm^2^	1.0	21.9	0.758	16.6	[159]
CuI	CuI/MAPbI_3_/PCBM/Al	0.06 cm^2^	1.04	21.6	0.62	13.58	[160]
	CuI/MAPbI_3_/C_60_/BCP/Ag	>1 cm^2^	0.97	22.7	0.738	16.8	[161]
	CuI/MAPbI_3_/PCBM/PEI/Ag	0.05 cm^2^	1.04	20.9	0.68	14.7	[162]
	PTAA−CuSCN/MAPbI_3_/PCBM/BCP/Ag	0.10 cm^2^	1.13	22.3	0.789	19.9	[163]
Cu_x_O	CuO_x_/MAPbI_3_/PCBM/Ag	~	0.952	17.5	0.662	11.0	[164]
	CuO_x_/MAPbI_3_/PCBM/ZnO/Al	5.5 mm^2^	1.03	22.42	0.76	17.43	[165]
CuS	CuS/MAPbI_3_/C_60_/BCP/Ag	0.10 cm^2^	1.02	22.3	0.712	16.2	[166]
Cu_3_PS_4_	Cu_3_PS_4_/MA_0.7_FA_0.3_PbI_3_/ PCBM/BCP/Ag	0.08 cm^2^	1.069	20.83	0.816	18.17	[167]
Modification	CuI/CuSCN/MAPbI_3−x_Cl_x_/PC_61_BM/PEI/Ag	0.09 cm^2^	1.11	22.33	0.76	18.76	[168]
	PTAA/CuI/FA_0.05_MA_0.95_PbI_3_/PCBM/C_60_−N/Ag	5.2 mm^2^	1.057	24.8	0.774	20.34	[169]

**Table 16 ijms-23-11792-t016:** Performances of several representative inverted PSCs on the basis of other inorganic semiconductor materials.

	Structure of PSCs	Area	V_oc_ (V)	J_sc_ (mA/cm^2^)	FF	PCE (%)	Ref.
GO	GO/MAPbI_3−x_Cl_x_/PCBM/ZnO/Al	7.25 mm^2^	1.0	17.46	0.71	12.4	[170]
	rGO/MAPbI_3_/PC_61_BM/BCP/Ag	0.09 cm^2^	0.98	15.4	0.716	10.8	[171]
	oxo−G_1_/MAPbI_3_/PCBM/ZnO/Al	~	1.08	18.06	0.777	15.2	[172]
	a−GO/MAPbI_3−x_Cl_x_/PCBM/BCP/Ag	4 mm^2^	1.0	18.4	0.768	14.14	[173]
V_2_O_5_	V_2_O_5_/PEDOT/MAPbI_3−x_Cl_x_/PCBM/ RhB101/LiF/Ag	0.11 cm^2^	1.05	13.59	0.59	8.4	[175]
	V_2_O_5_/P3CT−K/MAPbI_3_/PC_61_BM/ ZnO/Ag	0.06 cm^2^	1.09	23.24	0.779	19.7	[176]
MoO_x_	MoO_x_/MAPbI_3_/PCBM/Ag	~	0.99	18.8	0.71	13.1	[177]
CoO_x_	CoO_x_/MAPbI_3_/PCBM/Ag	0.02 cm^2^	0.95	20.28	0.755	14.5	[178]

**Table 17 ijms-23-11792-t017:** Performances of several representative inverted PSCs on the basis of C_60_ and C_60_−derivatives ETL.

	Structure of PSCs	Area	V_oc_ (V)	J_sc_ (mA/cm^2^)	FF	PCE (%)	Ref.
Pure PCBM	PEDOT:PSS/MAPbI_3−x_Cl_x_/PC_61_BM/Al	0.1 cm^2^	0.87	18.5	0.72	11.5	[193]
	PEDOT:PSS/MAPbI_3_/ PCBM/Al	0.1 cm^2^	1.05	19.98	0.78	16.3	[194]
Interface modification	NiO_x_/NiO_nc_/MAPbI_3_/ PCBM/BCP/Al	0.06 cm^2^	1.04	13.2	0.69	9.51	[195]
	P3CT−Na/MAPbI_3_/PCBM/C_60_/Al	~	1.07	21.1	0.73	16.6	[196]
	PTAA/MAPbI_3_/PCBM/C_60_/BCP/Al	7.25 mm^2^	1.07	22.0	0.768	18.1	[89]
	Poly−TPD/MAPbI_3_/PCBM/LiF/Al	0.045 cm^2^	1.07	21.8	0.737	17.2	[118]
	Li_0.05_Mg_0.15_Ni_0.8_O/MAPbI_3_/PCBM/Ti(Nb)Ox/Ag	1.02 cm^2^	1.083	20.4	0.827	18.3	[147]
	PEDOT:PSS/MAPbI_3_/ PCBM/π−PFEs/Ag	~	1.04	22.11	0.799	18.46	[197]
Doping modification	PEDOT:PSS/MAPbI_3_/ PCBM:GD/C_60_/Al	0.06 cm^2^	0.969	23.4	0.654	14.8	[198]
	PEDOT:PSS/MAPbI_3−x_Cl_x_/Oleamide:PCBM/Ag	14.0 mm^2^	0.98	18.76	0.693	12.69	[199]
	NiMgLiO/MAPbI_3_/H3:PCBM/BCP/Ag	~	1.06	21.5	0.793	18.1	[200]
	NiO/MAPbI_3_/PCBM:CoSe/Ag	0.07 cm^2^	1.073	19.85	0.70	14.91	[201]
	PEDOT:PSS/MAPbI_3_/rGO:PCBM/Ag	4 mm^2^	0.942	23.52	0.655	14.51	[202]
	P3CT−K/MAPbI_3_/PC_61_BM:SnO_2_/Ag	0.06 cm^2^	1.12	23.15	0.76	19.7	[203]
	PEDOT:PSS/MAPbI_3_/ PCBM:F8TBT/Ag	6.25 cm^2^	1.12	22.43	0.82	20.6	[204]

**Table 18 ijms-23-11792-t018:** Performances of several representative inverted PSCs on the basis of organic small−molecule materials ETL.

Structure of PSCs	Area	V_oc_ (V)	J_sc_ (mA/cm^2^)	FF	PCE (%)	Ref.
PEDOT:PSS/MAPbI_3_/Br−PDI/ZnO/Ag	0.12 cm^2^	0.83	18.9	0.669	10.5	[207]
PEDOT:PSS/MAPbI_3_/TDTP/LiF/Ag	~	1.05	22.4	0.777	18.2	[208]
P3CT−Na/MAPbI_3_/TPE−PDI4/ Rhodamine 101/LiF/Ag	~	1.052	21.98	0.81	18.78	[209]
P3CT−Na/MAPbI_3_/ITCPTC−Th/ Rhodamine 101/LiF/Ag	~	1.029	21.77	0.76	17.11	[210]

**Table 19 ijms-23-11792-t019:** Performances of several representative inverted PSCs on the basis of inorganic ETL.

Structure of PSCs	Area	V_oc_ (V)	J_sc_ (mA/cm^2^)	FF	PCE (%)	Ref.
NiO_x_/MAPbI_3_/ZnO/Al	0.1 cm^2^	1.01	21.0	0.760	16.1	[212]
NiO/MAPbI_3_/Bi_2_S_3_/Au	~	0.949	18.6	0.742	13.1	[213]
PEDOT:PSS/MAPbI_3_/ CdSe QDs/LiF/Ag	0.04 cm^2^	0.985	21.8	0.703	15.1	[214]
NiO_x_/MAPbI_3_/CeO_x_/Ag	~	1.047	20.43	0.797	17.1	[215]
NiMgLiO/CsPbI_2_Br/C−TiO_2_/ Bi/Ag	0.09 cm^2^	1.26	14.72	0.76	14.0	[216]
Cu_2_O/MAPbI_3_/SiO_2_/GZO/Ag	0.1 cm^2^	1.12	20.9	0.786	18.4	[217]
NiO_x_/P1:FA_0.85_MA_0.15_PbI_2.55_Br_0.45_/ Nb_2_O_5_/Ag	0.16 cm^2^	1.098	23.59	0.788	20.41	[218]
NiO_x_/MAPbI_3_/Sn:In_2_O_3_/In_2_O_3_/Ag	~	1.10	23.22	0.809	20.65	[219]
c−NiO_x_/mp−NiO_x_/ Cs_0.05_(FA_0.83_MA_0.17_Pb(I_0.83_Br_0.17_))_0.95_/ IZO/Al	0.2 cm^2^	1.02	22.86	0.694	16.2	[220]
Cu:NiO_x_/MAPbI_3_/CdS/Au	~	1.012	19.74	0.668	13.36	[221]
MoS_2_/MAPbI_3_/TiO_2_/Ag	~	0.93	26.24	0.83	20.43	[222]
NiMgLiO/CsPbI_2_Br/C−TiO_2_/Sb	0.09 cm^2^	1.28	15.0	0.77	14.8	[223]
Cu:NiO_x_/MAPbI_3_/ZnO/Ag	0.2 cm^2^	1.03	23.05	0.64	16.51	[224]
Mo/Cu_2_ZnSnSe_4_/MAPbI_3_/ZnS/ IZO/Ag	0.1 cm^2^	1.1	20.8	0.763	17.4	[225]
NiO_x_/Cs_0.05_FA_0.79_MA_0.16_PbBr_0.51_I_2.49_/ SnO_x_/Al	0.126 cm^2^	1.12	18.1	0.661	13.5	[226]

**Table 20 ijms-23-11792-t020:** Performances of several representative all−inorganic inverted PSCs.

Structure of PSCs	Area	V_oc_ (V)	J_sc_ (mA/cm^2^)	FF	PCE (%)	Ref.
NiO/CsPbI_3_/C_60_/ZnO/Ag	~	1.19	14.25	0.776	13.1	[229]
NiO_x_/CsPbIBr_2_/MoO_x_/Au	9 mm^2^	0.85	10.56	0.62	5.52	[230]
Cs−NiO_x_/N749/CsPbIBr_2_/ PCBM/BCP/Ag	~	1.19	11.49	0.69	9.49	[231]
NiO_x_/CsPbI_2_Br/ZnO@C_60_/Ag	~	1.14	15.2	0.77	13.1	[232]
NiMgLiO/CsPbI_2_Br/ C−TiO_2_/Sb	0.09 cm^2^	1.28	15.0	0.77	14.8	[223]
NiO_x_/CsPbI_2_Br/ZnO@C_60_/Ag	0.09 cm^2^	1.1	15.1	0.756	12.6	[233]
NiO_x_/CsPbI_2_Br/PCBM/Ag	~	1.12	15.6	0.761	13.3	[234]
NiO_x_/CsPbI_2_Br/ZnO/C_60_/Ag	0.09 cm^2^	1.15	16.12	0.776	14.38	[235]
NiO_x_/Zn:CuGaO_2_/F−CsPbI_2_Br/TiO_2_/PC_61_BM/ZnO	0.09 cm^2^	1.17	15.98	0.807	15.1	[236]
NiO_x_/CsPbI_2_Br/6TIC−4F/ ZnO/C_60_/Ag	0.0672 cm^2^	1.16	17.7	0.786	16.1	[237]

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
