# Peer review of "Recent Advances in Inverted Perovskite Solar Cells: Designing and Fabrication"

_ijms, 2022, doi:10.3390/ijms231911792_

Round 1
Reviewer 1 Report
The authors have described a review about recent advances in inverted perovskite solar cells. They introduced the design and fabrication of inverted devices. The advantages and disadvantages are analyzed in detail, and the optimization methods are summarized by the authorships. I think it can make a contribution to the development of inverted perovskite solar cells, and which can give help to the researchers in the field. So, I recommend the manuscript for publication after minor revisions.
(1) Good work, but with few problems in English usage. The syntax can be further refined.
(2) Could the authors give us more discussion on the design and fabrication of perovskite absorption layer?
(3) There is no reference after Table 1 (Line 168), Table 2 (Line 195), Table 3 (Line 240), etc. The references should be showed after the “tables” appearing in the text. Please revise them.
(4) The name in Line 579 (Wenxiu Que) is not consistent with the name of author before, which only showed the last name. Please check the full text carefully and correct them.
(5) In addition, the quality of the figures should be improved, which are not clear enough, such as figure 1 and 2.

Reviewer 2 Report
In this review article, the author categorizes the references on inverted perovskite solar cells according to different ETL and HTL materials, which can help readers review the technology more quickly. Here are few questions and suggestions.
1. In addition to the issue of flexible substrates, inverted perovskite solar cells are also more suitable for 2T perovskite-silicon tandem solar cells than the traditional upright perovskite solar cells. It is recommended that the author provide additional clarifications.
2. When comparing the efficiency of perovskite solar cells in different reference, it is recommended to add the device area to the table in order to compare the differences more comprehensively and objectively.
3. Since large area and stability are one of the bottlenecks of perovskite solar cells toward commercialization, it is recommended that the authors present the large area and stability performance of current inverted perovskite solar cells, or discuss the area and stability performance difference between inverted and traditional upright.
Reviewer 3 Report
The authors compiled a very comprehensive review of recent advances in inverted structure solar cells based on perovskites. The reports were logically categorized and tabulated according to the hole or electron transport layer and substance. A general proofreading is suggested as the text could benefit in many places, some of them highlighted in the attached pdf file. There are also several additional reports in the literature to make for a comprehensive review and expand the scope for development of more efficient perovskite solar cells.
The review is well organized, nevertheless the manuscript can benefit from a general proofreading. The following references published by the reviewer were suggested in line with the text fo the manuscript:
Page 21, line 747: Encapsulation can be also achieved with down-shifting QDs that enables filtering of UV radiation with increased solar cell stability as reported in https://doi.org/10.1016/j.orgel.2017.06.056
Such photoluminescent materials can be directly added to stabilize and monitor the performance of absorber, electron or hole transport layers and electrodes of a solar cell, as reported in
https://doi.org/10.1002/solr.201900270
and
https://doi.org/10.1016/j.solener.2019.11.050
. A discussion of these advancements is currently missing from the manuscript, but is required for a comprehensive review.
